# Nitric oxide controls shoot meristem activity via regulation of DNA methylation

Jian Zeng[1,4], Xin'Ai Zhao[1,4], Zhe Liang [1,2], Inés Hidalgo [1], Michael Gebert [1,3], Pengfei Fan[1], Christian Wenzl[1], Sebastian G. Gornik[1] & Jan U. Lohmann [1]✉

Despite the importance of Nitric Oxide (NO) as signaling molecule in both plant and animal development, the regulatory mechanisms downstream of NO remain largely unclear. Here, we show that NO is involved in *Arabidopsis* shoot stem cell control via modifying expression and activity of *ARGONAUTE 4* (*AGO4*), a core component of the RNA-directed DNA Methylation (RdDM) pathway. Mutations in components of the RdDM pathway cause meristematic defects, and reduce responses of the stem cell system to NO signaling. Importantly, we find that the stem cell inducing WUSCHEL transcription factor directly interacts with AGO4 in a NO dependent manner, explaining how these two signaling systems may converge to modify DNA methylation patterns. Taken together, our results reveal that NO signaling plays an important role in controlling plant stem cell homeostasis via the regulation of de novo DNA methylation.

Growth and development of plants rely on the continuous division and differentiation of stem cells that reside in specialized tissues, called meristems, located at the growth points of shoot and root, respectively. Stem cell fate in the shoot apical meristem (SAM) of the reference plant *Arabidopsis thaliana* is tightly controlled by a complex regulatory network, including transcription factors and multiple signaling molecules[1–3]. A negative feedback loop between the stem cell inducing WUSCHEL (WUS) homeodomain transcription factor and the secreted peptide ligand CLAVATA3 (CLV3) is essential to maintain stem cell fate in the SAM[4,5]. *WUS* mRNA is expressed in the organizing center (OC), however, WUS protein moves apically to the stem cells where it induces stem cell fate and activates *CLV3* expression[6,7]. Conversely, stem cells secrete the small peptide CLV3 that represses WUS expression in the niche via the CLAVATA1/2 (CLV1/2) and CORYNE (CRN) receptors, thus limiting the number of stem cells[4,5,8,9]. The peripheral zone (PZ) is found laterally to stem and niche cells and represents a domain of rapid cell proliferation, as well as for specification of organ primordia[1–3,10–12]. How stem cell activity is coordinated with cell proliferation and differentiation at the periphery is not well understood, but recent work has pointed to an important role of the CLE40 peptide acting via the BAM1 receptor in the communication of PZ and CZ[13].

Nitric oxide (NO) is a gaseous biomolecule of the class of reactive nitrogen species (RNS), and has emerged as an important signaling molecule involved in diverse biological processes in both plants and animals[14–20]. In plants, NO has been shown to play a role in developmental and physiological processes, including floral transition, reaction to hypoxia, as well as immune and defense responses[17,21–23]. Additionally, several lines of evidence have suggested that redox state plays an important role in stem cell homeostasis, including the key redox component RNS and reactive oxygen species (ROS)[15,19,24–26]. NO and ROS are involved in controlling differentiation of embryonic stem cells in animals[15,27,28] and more recently, they have been shown to act as important signals in root and shoot stem cell systems of plants[16,19,20,24,25].

However, little is known about the molecular mechanisms underlying NO activity in the control of stem cell fate in plants. In this study, we show that NO acts via the ARGONAUTE protein AGO4 at the transcriptional and post-translational level to affect genome wide DNA methylation patterns, which in turn are essential for proper meristem function.

[1]Department of Stem Cell Biology, Centre for Organismal Studies, Heidelberg University, 69120 Heidelberg, Germany. [2]Present address: Biotechnology Research Institute, Chinese Academy of Agricultural Sciences, Beijing 100081, China. [3]Present address: CureVac, 72076 Tübingen, Germany. [4]These authors contributed equally: Jian Zeng, Xin'Ai Zhao. ✉e-mail: jan.lohmann@cos.uni-heidelberg.de

## Results

### Nitric oxide promotes peripheral zone fate via repression of WUS

To investigate the role of NO in plant stem cell regulation, we first examined NO distribution in the SAM using a sensitive fluorescent indicator (4,5-diaminofluorescein diacetate, DAF-2DA)[17,29]. We found that NO accumulated to higher levels in the peripheral zone (PZ) compared to the central zone (CZ) (Supplementary Fig. 1a, d), suggesting that NO could act as a signal to define PZ cell fate. To explore the mechanism behind NO accumulation in the SAM periphery, we next examined the expression patterns of three important NO biosynthesis genes by in situ hybridization, including *NIA1*, *NIA2,* and *NOA1*[17,20,29–32]. Interestingly, mRNAs for all three genes specifically accumulated in the PZ and cells of organ boundaries, but were absent from stem cells (Supplementary Fig. 2), consistent with NO distribution.

To functionally test the contribution of NO biosynthesis genes to SAM function, we analyzed NO-deficient mutants. At the seedling stage, both *noa1* single mutants, as well as *nia1/nia2/noa1* triple mutants exhibited a substantial delay in generating the first pair of true leaves (Fig. 1a, b, Supplementary Fig. 3), while after bolting they produced fewer floral buds compared to wild-type (Fig. 1c, d). Notably, SAMs of *nia1/nia2/noa1* triple mutants were significantly smaller than those of wild-type controls, both at the seedling and inflorescence stage (Fig. 1m). To test whether these phenotypes were due to reduced stem cell activity or reduction of the PZ, we analyzed the expression of *WUS* and *CLV3* as markers for organizing cells and stem cells, respectively, which are both found in the center of the SAM, as well as *UFO* and *CUC2* as markers for the PZ and boundaries. Consistent with a role of NO in promoting PZ cell fate, we found that the expression of the stem cell regulator *WUS* was significantly increased in *nia1/nia2/noa1* triple mutants, at both the vegetative and

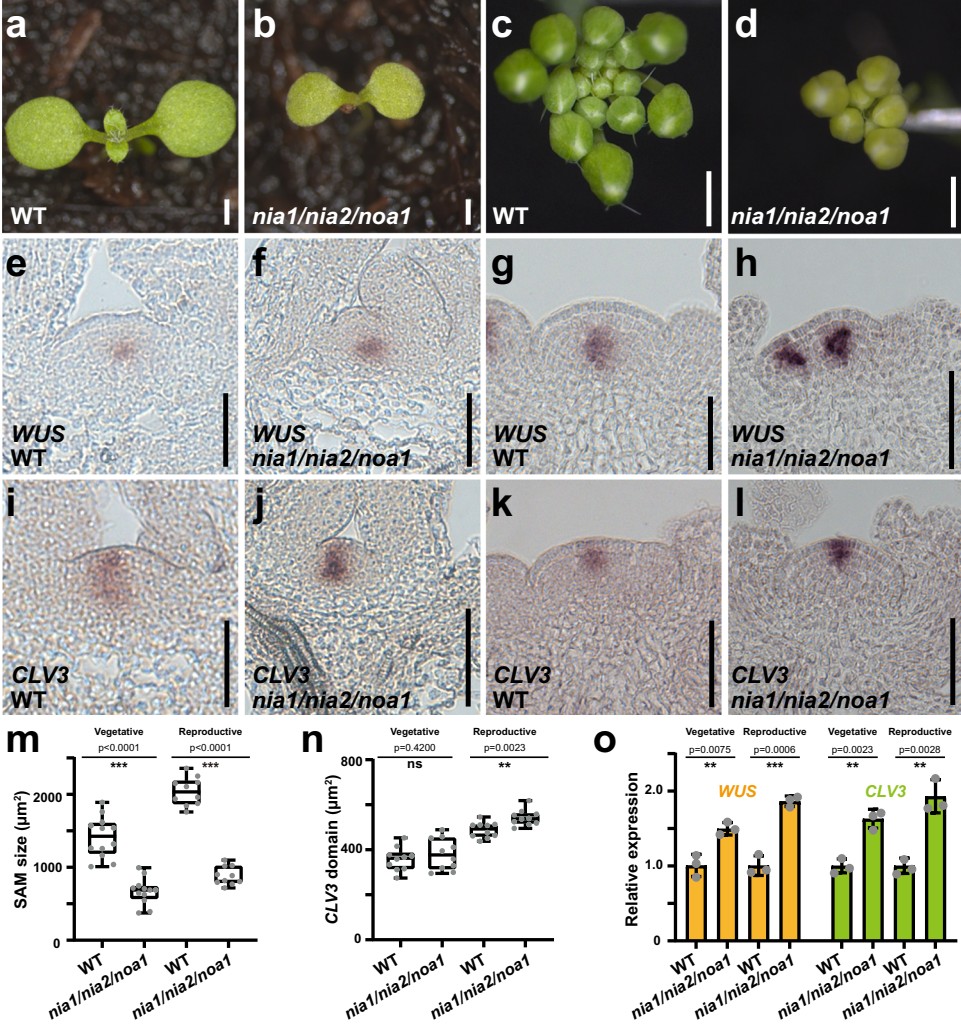

**Fig. 1 | Nitric oxide is required for proper meristem activity. a–d** Representative phenotypes of wild-type (**a**) and *nia1/nia2/noa1* mutant (**b**) seedlings seven days after germination, as well as wild-type (**c**) and *nia1/nia2/noa1* mutants (**d**) during inflorescence stage. Scale bars: 1 mm. **e–h** *WUS* expression patterns in wild-type (**e**) and *nia1/nia2/noa1* mutant (**f**) seedlings, as well as wild-type (**g**) and *nia1/nia2/noa1* mutants (**h**) during inflorescence stage. Scale bars, 50 μm. **i–l** *CLV3* expression patterns in wild-type (**i**) and *nia1/nia2/noa1* mutant (**j**) seedlings, as well as wild-type (**k**) and *nia1/nia2/noa1* mutants (**l**) during inflorescence stage. Scale bars, 50 μm. **m**, **n** Quantification of SAM size (**m**) and the size of the *CLV3* positive stem cell domain (**n**) in wild-type and *nia1/nia2/noa1* mutants at both the vegetative (*n* = 12 and *n* = 10 independent plants were used for quantification of the SAM size and the size of the *CLV3* expression domain, respectively.) and reproductive (*n* = 10 independent plants) stages. The error bars indicate the highest and lowest values, the box indicates the middle 50%, the center line indicates the median, the whiskers indicate the data range within 1.5× the interquartile range, and outliers are not shown. Two-sided Student's *t*-tests, **\*\*p* < 0.01, \*\*\**p* < 0.001; ns no significant difference. **o** RT-qPCR quantifications of *WUS* and *CLV3* expression levels in wild-type and the *nia1/nia2/noa1* mutants shown as mean ± s.d.; *n* = 3 biological replicates, two-sided Student's *t*-tests, **\*\*p* < 0.01, \*\*\**p* < 0.001. All experiments were repeated independently at least two times on pools of apices of at least 30 plants, with similar results.

reproductive stages (Fig. 1e–h, o), along with an expansion of the stem cell domain during the reproductive stage and an increase of *CLV3* expression (Fig. 1i–l, n, o). Importantly, we found that peripheral *UFO* expression was substantially reduced in triple mutants compared to wild-type, while the central expression was unaffected (Supplementary Fig. 4a, b). Similarly, we observed that the wild-type peripheral *CUC2* expression domain was lost and replaced by a much more central RNA accumulation pattern in the triple mutants (Supplementary Fig. 4h, i). Taken together, these results suggested that NO acts in the periphery of the SAM to promote PZ cell fate, which is marked by high cell proliferation. At the same time, NO signaling represses central cell fate, likely via repressing *WUS* expression.

To circumvent the complex feedbacks of SAM regulation and test the short-term effects of NO on stem cell regulation, we transiently reduced endogenous NO levels using ethanol-inducible expression of an artificial microRNA against *NOA1*. After induction by 1% ethanol for 12 h, *NOA1* expression levels were significantly reduced in transgenic plants (Supplementary Fig. 1h), along with endogenous NO levels (Supplementary Fig. 1b, c, g). In contrast, endogenous NO levels were not changed in the *p35S::AlcR; pAlcA::GUS* control plants (Supplementary Fig. 1e–g). Consistent with our results using stable mutant alleles, we observed an increase in both *WUS* and *CLV3* expression following the reduction of endogenous NO (Supplementary Fig. 1h). To further explore the direct effects of NO signaling in the SAM, we turned to a pharmacological approach using a transgenic double reporter line for *WUS* and *CLV3* and 3 h treatments with the NO donor sodium nitroprusside (SNP)[33] and the NO scavenger 2-4-carboxyphenyl-4,4,5,5-tetramethylimidazoline-1-oxyl-3-oxide (c-PTIO)[34]. Consistent with our genetic interference with NO biosynthesis, we found the *WUS* reporter signal to be significantly reduced following activation of NO by SNP treatment (Fig. 2b, m). Conversely, *WUS* reporter levels and domain were dramatically increased after reduction of endogenous NO levels by c-PTIO treatment (Fig. 2c, m). Importantly, GFP fluorescence was not affected by modifying NO levels (Supplementary Fig. 5a–d). Interestingly, the *CLV3* reporter was not significantly changed following either treatment (Fig. 2e, f, k, l), supporting the notion that NO acts mainly via *WUS* and that the indirect effect on stem cells had not set in after 3 h of perturbation. To test this hypothesis directly, we continuously increased NO levels in the CZ by expressing *NOA1*, one of the NO biosynthesis gene, from the *CLV3* promoter. Supporting the idea that NO acts on stem cells indirectly via *WUS*, we observed a reduction of both *WUS* and *CLV3* expression levels in the transgenic plants (Supplementary Fig. 6a–e). Consistent with an expansion of the PZ and associated enhanced cell proliferation, the transgenic plants had enlarged SAMs, despite the reduction in the *CLV3* expression domain (Supplementary Fig. 6f, g). Taken together, our genetic and pharmacological results showed that NO acts from the periphery to promote PZ identity by limiting *WUS* expression and in turn stem cell fate.

One important mechanism required for proper SAM function in this scenario is the repression of NO biosynthesis in the cells of the organizing center and the stem cells. Since WUS has the capacity to act as a transcriptional repressor and is active in these two cell populations[35–37], we inspected our resources on direct WUS target genes. Indeed, two of the NO biosynthesis genes, *NIA1* and *NIA2*, were previously identified by ChIP-seq[38] (Supplementary Fig. 7). We further tested the transcriptional response of *NIA1*, *NIA2*, and *NOA1* to ectopically induced WUS activity and found that at the whole seedling level *NIA1* and *NOA1* showed a robust repression after 4 h of WUS induction (Supplementary Fig. 8a–c). Consistent with our published data, we detected WUS binding to the promoter of *NIA1* by ChIP-PCR in a region containing putative WUS binding sites (Supplementary Fig. 8d). Electrophoretic mobility shift assays (EMSAs) confirmed that WUS was able to bind to the ChIP-positive fragment in vitro (Supplementary Fig. 8e). These results indicated that WUS has the potential to directly repress *NIA1* expression and suggested that the exclusion of NO biosynthesis

genes from the CZ could be dependent on this stem cell master regulator.

## Nitric oxide is a key regulator of ARGONAUTE 4

To elucidate the mechanisms underlying the PZ cell fate promoting effect of NO, we next characterized the global transcriptional response to the NO donor SNP specifically in meristems. Previously, NO response genes were only identified in whole seedlings, severely limiting the detection of meristematic transcripts[25]. To circumvent this problem, we took advantage of the *ap1/cal* mutant, which is characterized by a massive over-proliferation and subsequent arrest of inflorescence and young floral meristems. To first test whether the global NO response in *ap1/cal* mutants is comparable to wild-type, we performed mRNA sequencing on wild-type and *ap1/cal* seedlings with and without SNP treatments. We observed a total of 10,993 and 12,401 differentially expressed genes (DEGs) for wild-type and *ap1/cal* mutants, respectively. Importantly, 88% DEGs identified in the wild-type were confirmed in *ap1/cal* mutants (Supplementary Fig. 9), suggesting that the NO response in both genotypes is very similar. To identify meristem specific NO response genes, we next applied RNA-seq to micro-dissected shoot apices of *ap1/cal* mutants, which were either SNP or mock-treated for 24 h. From the data, we were able to identify 3217 DEGs, including 1309 mRNAs with reduced expression and 1908 transcripts showing elevated RNA accumulation (Supplementary Data 1 and 2). Zooming in on genes that may mediate the repression of stem cell fate, we integrated the set of SNP repressed transcripts with three published resources representing genes with expression the center of the SAM on the one hand[39,40] and proteins which can be S-nitrosylated on the other hand[41]. S-nitrosylation is one of the most important post-translational modification mechanisms for NO function[41–44], and thus, this integration allowed us to focus on targets which were co-expressed with *WUS* and *CLV3* and could be regulated by NO at both transcript and protein levels. Interestingly, we only obtained one gene that was present in all four datasets (Fig. 3a, Supplementary Data 1–3), namely *ARGONAUTE 4 (AGO4)*. *AGO4* is known to function in the RNA-directed DNA methylation (RdDM) pathway and is most closely related to *AGO6* that had been shown to be expressed in the shoot meristem[45–49]. This finding led us to hypothesize that *AGO4* might be involved in transmitting NO signals in the SAM via regulation of DNA methylation. To test this hypothesis, we first characterized the expression pattern of *AGO4* in the SAM. By in situ hybridization, we found that *AGO4* mRNA specifically accumulates in stem cells and niche cells, and is low or absent in cells at the periphery of the SAM in a pattern complementary to NO distribution (Fig. 3b–d, Supplementary Fig. 10). This finding was consistent with *AGO4* expression being reduced by treatment with the NO donor SNP in our RNA-seq experiment (Fig. 3k) and suggested that NO signaling in the PZ could repress *AGO4* expression. To verify these findings with spatial resolution, we treated plants with the NO donor SNP and examined *AGO4* expression by in situ hybridization. Indeed, *AGO4* mRNA levels were significantly reduced after SNP treatment in both wild-type and *ap1/cal* mutants (Fig. 3e–h), demonstrating that elevated NO signaling is sufficient to repress *AGO4* expression even in stem cells and niche cells. To test if endogenous NO is required to define the specific *AGO4* expression pattern, we analyzed SAMs of *nia1/nia2/noa1* triple mutants and observed that *AGO4* expression was substantially increased and expanded into the periphery of the SAM (Fig. 3i, j, l). Taken together, these results suggested that NO signaling is an essential upstream regulator of *AGO4* transcription and acts to limit its expression to the CZ.

Since published data showed that AGO4 protein can be S-nitrosylated at C482[41], we next examined whether NO affects AGO4 also at the post-transcriptional level. To this end, we performed a biotin switch assay[50] using WT AGO4 (AGO4$^{WT}$) and mutated AGO4 protein (AGO4$^{C482S}$) and found that both forms of the AGO4 could be

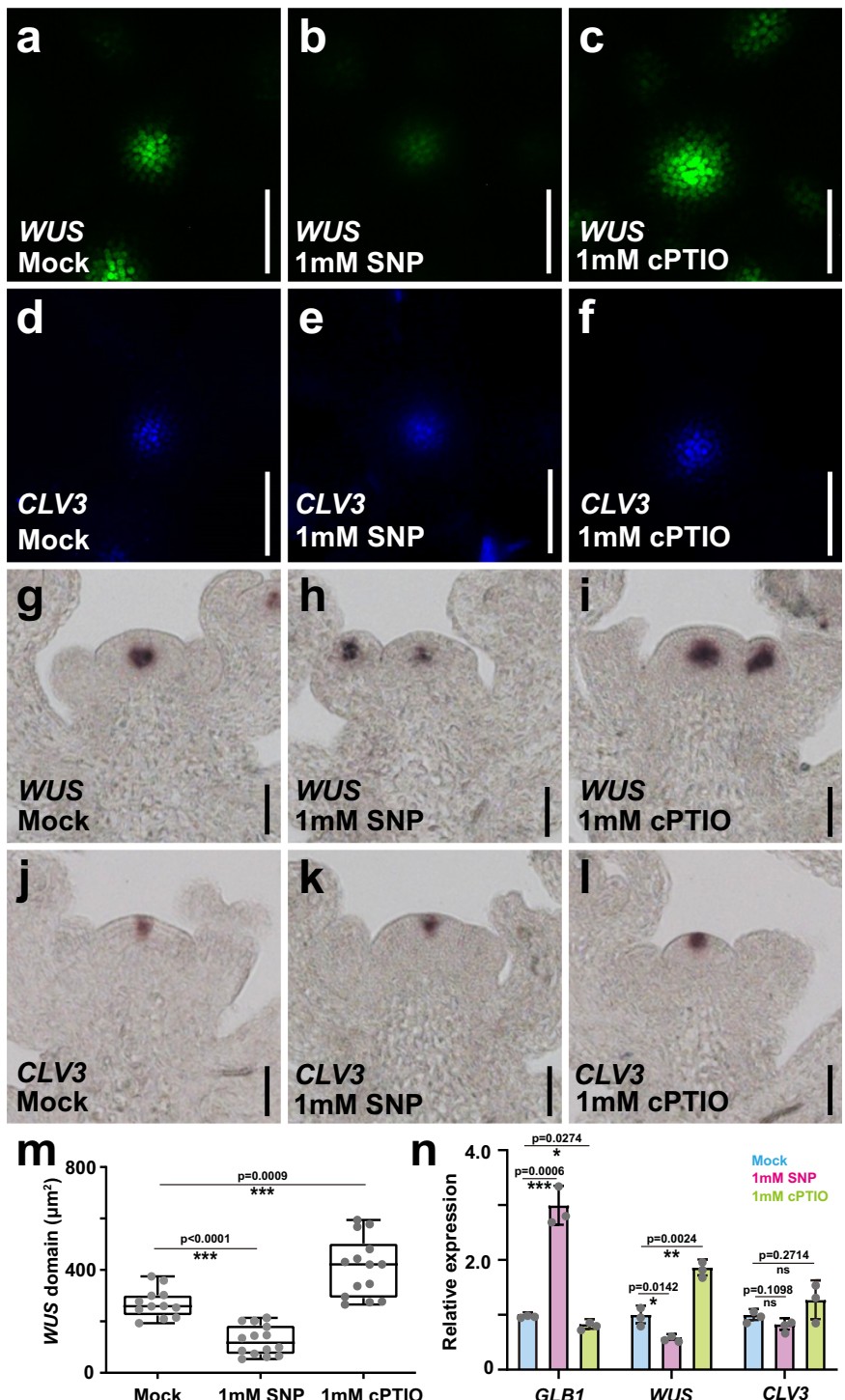

**Fig. 2 | Nitric oxide limits *WUS* expression. a–f** Short-term responses of stem or niche cell reporters to increased or reduced NO levels. *WUS* (**a**–**c**) and *CLV3* (**d**–**f**) reporters after 3 h of mock (**a**, **d**), SNP (**b**, **e**) and c-PTIO treatments (**c**, **f**). Scale bars, 50 μm. **g–l** Short-term responses of endogenous stem or niche cell marker transcripts to increased or reduced NO levels. Detection of *WUS* (**g**–**i**) and *CLV3* (**j**–**l**) expression patterns after 3 h of mock (**g**, **j**), SNP (**h**, **k**), and c-PTIO treatments (**i**, **l**) by in situ hybridization. Scale bars, 50 μm. **m** Quantification of the *WUS* expression domain after 3 h of mock (*n* = 12 independent plants), SNP (*n* = 14 independent plants), and c-PTIO (*n* = 14 independent plants) treatments. The error bars indicate the highest and lowest values, the box indicates the middle 50%, the center line indicates the median, the whiskers indicate the data range within 1.5× the interquartile range, and outliers are not shown. Two-sided Student's t-tests, ***$P < 0.001$. **n** RT-qPCR quantification of *GLB1*, *WUS,* and *CLV3* expression levels after 3 h of mock, SNP, and c-PTIO treatments shown as mean ± s.d.; *n* = 3 biological replicates, two-sided Student's t-tests, *$p < 0.05$, **$p < 0.01$, ***$p < 0.001$; ns, no significant difference. All experiments were repeated independently at least two times on pools of apices of at least 30 plants, with similar results.

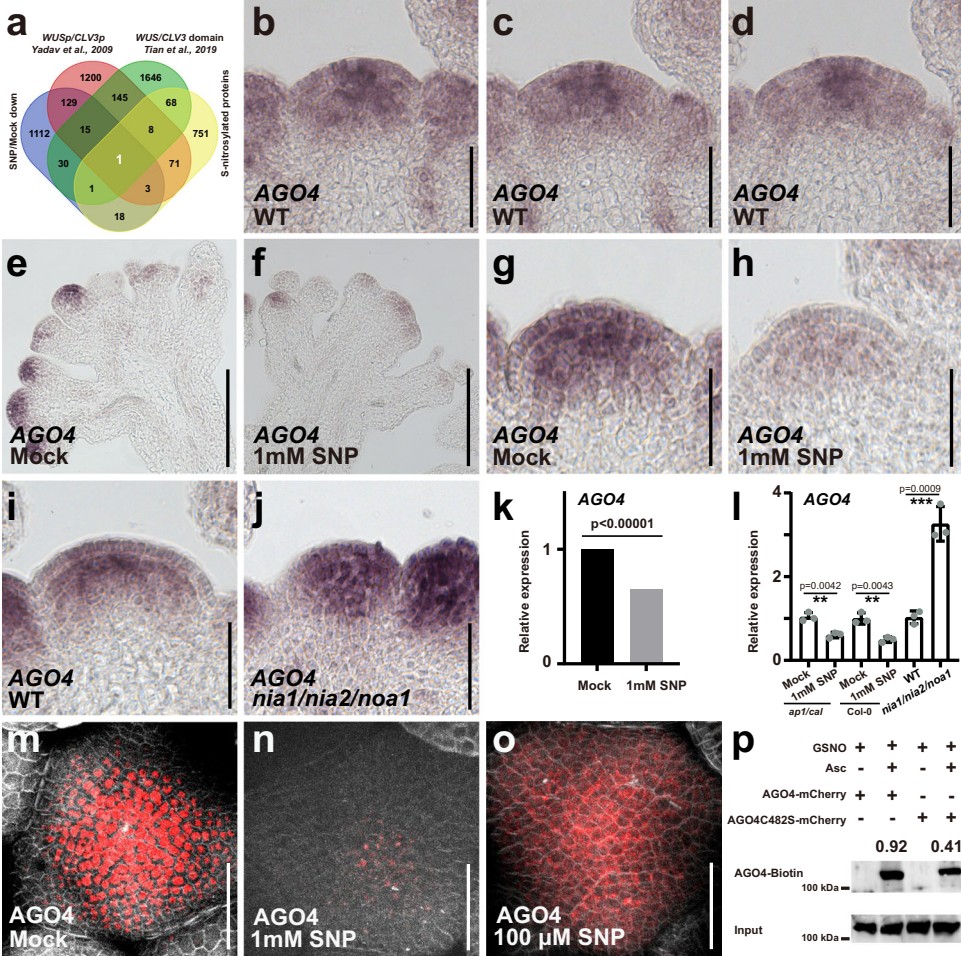

**Fig. 3 | Nitric oxide limits *AGO4* accumulation to the center of the SAM. a** Venn diagram showing the overlap of stem cell and niche cell specific transcripts (Tian et al.[39], Yadav et al.[40]), genes with reduced expression following SNP treatment and S-nitrosylated proteins (Hu et al.[41]). **b–d** Serial longitudinal sections through a SAM probed for *AGO4* mRNA. Scale bars, 50 μm. **e, f** *AGO4* expression patterns in shoot apices of *ap1/cal* mutants after 3 h of mock (**e**) or SNP treatment (**f**). Scale bars, 50 μm. **g, h** *AGO4* expression patterns in shoot apices of wild-type plants after 3 h of mock (**g**) or SNP treatment (**h**). Scale bars, 50 μm. **i, j** *AGO4* expression patterns in wild-type (**i**) and *nia1/nia2/noa1* mutants (**j**). Scale bars, 50 μm. **k** *AGO4* transcript accumulation following SNP treatment as quantified by RNA-seq. **l** Quantification of *AGO4* expression levels in *ap1/cal* and wild-type plants after SNP treatment, and *nia1/nia2/noa1* mutants by RT-qPCR shown as mean ± s.d.; *n* = 3 biological replicates, two-sided Student's t-tests, \*\**P* < 0.01, \*\*\**p* < 0.001. **m–o** SAMs of *pAGO4::mCherry-AGO4/ago4* rescue plants after 3 h of mock (**m**), 1 mM SNP (**n**) or 100 μM SNP (**o**) treatment. Scale bars, 50 μm. **p** Analysis of S-nitrosylation of AGO4 protein by in vivo biotin-switch method. Asc, ascorbate sodium. The experiments in **b–j** and **m–o** were repeated twice on pools of apices of at least 15 plants, and the experiments in (**p**) were repeated three times, with similar results.

modified by S-nitroslation in response to NO treatment. Interestingly, AGO4^C482S was less sensitive to the biotin switch (Fig. 3p), suggesting that C482 is a functional nitrosylation site, but that additional targets for this modification are likely to exist in AGO4. Since S-nitroslation can affect protein activity, sub-cellular localization, or protein stability[41–44], we investigated the response of AGO4 protein to NO in vivo. To this end, we treated mCherry-AGO4 rescue plants[51] with the NO donor SNP and found that following our standard treatment with 1 mM SNP mCherry-AGO4 signal was almost completely lost (Fig. 3m, n, Supplementary Fig. 11). Reducing SNP concentrations to 100 μM revealed that mCherry-AGO4 was found in the cytoplasm instead of accumulating in the nucleus as in control plants (Fig. 3m, o, Supplementary Fig. 11). Importantly, signal from mCherry-NLS controls was not affected by SNP (Supplementary Fig. 5e–h). Taken together, our results showed that NO is a key regulator of expression and activity of AGO4, which functions in the small RNA-directed DNA methylation pathway.

**AGO4 protein connects WUS function with the RdDM pathway**
*AGO4* is a key component of the small RNA-induced silencing pathway and can direct de novo DNA methylation in the RNA-directed DNA methylation (RdDM) pathway, together with its orthologues *AGO6* and

*AGO9*[45–49]. To functionally test the contribution of *AGO4* and the RdDM pathway to SAM activity, we first examined *ago4* single mutants, *ago4/ago6/ago9* triple mutants, as well as *drm1/drm2* mutants, *rdm1* mutants and *drm1/drm2/met1* triple mutants representing downstream components involved in DNA methylation. We observed that *ago4/ago6/ago9*, *drm1/drm2*, and *rdm1* mutants showed a reduced number of organs after flowering (Fig. 4a, c, Supplementary Fig. 12a–c), whereas *ago4* single mutants didn't show obvious defects (Fig. 4a, b). This suggested that *AGO4* likely acts redundantly with *AGO6* and *AGO9* in maintaining a proper SAM, and that the DNA methylation enzymes of the RdDM pathway share this function. To explore the molecular mechanisms of RdDM-mediated SAM regulation, we analyzed *WUS* and *CLV3* expression in SAMs of our set of mutants. Consistent with the observed phenotypes, we found that *WUS* transcripts were significantly reduced in *ago4/ago6/ago9* and *drm1/drm2/met1* triple mutants, whereas there was no obvious difference in *ago4* single mutants (Fig. 4e–h, o). Surprisingly, we observed increased expression of *CLV3* in both triple mutant lines, suggesting that inactivating the RdDM pathway makes *CLV3* expression partially independent from the activation by WUS (Fig. 4i–l, o). This notion was supported by the finding that while SAM size was significantly reduced in the RdDM

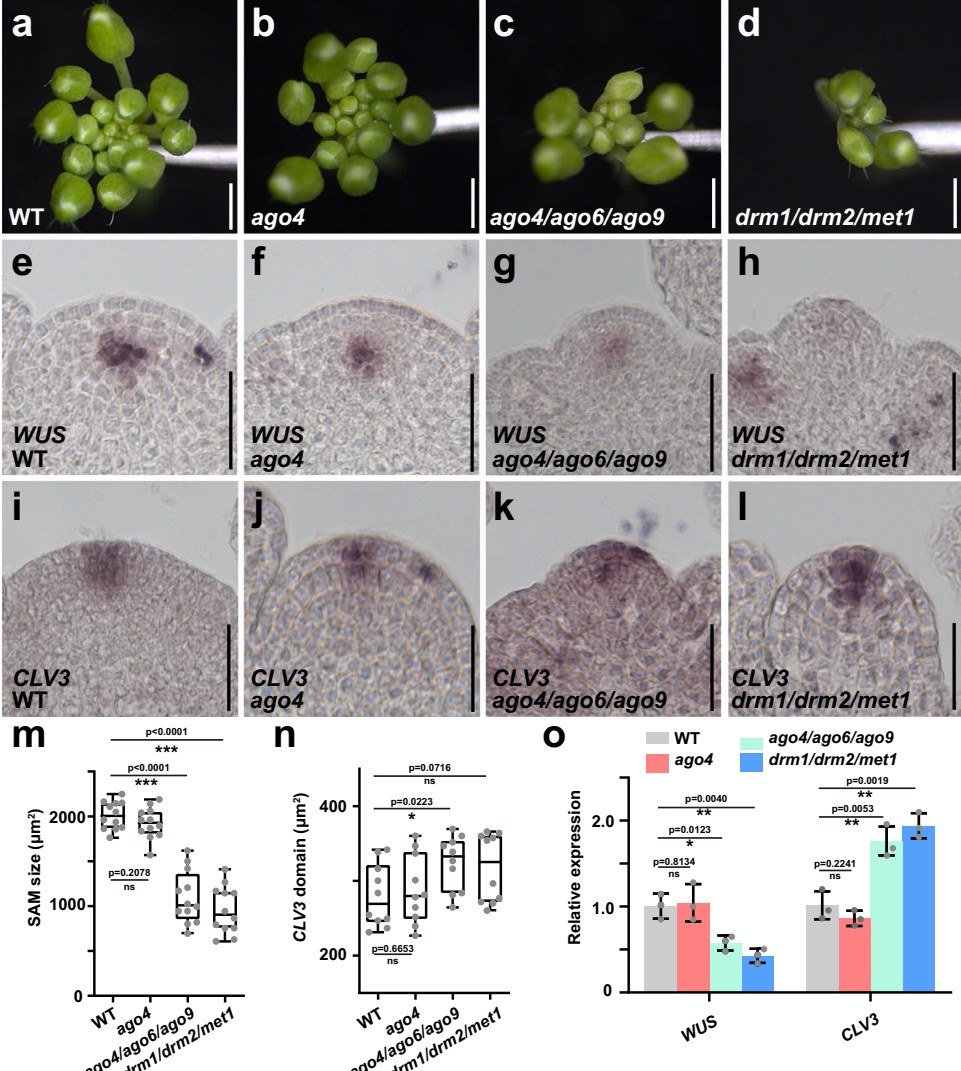

**Fig. 4 | The RdDM pathway is required for stem cell homeostasis.**
**a–d** Representative inflorescence phenotypes of wild-type (**a**), *ago4* (**b**), *ago4/ago6/ago9* (**c**), and *drm1/drm2/met1* mutants (**d**). Scale bars, 1 mm. **e–h** *WUS* expression patterns in wild-type (**e**), *ago4* (**f**), *ago4/ago6/ago9* (**g**) and *drm1/drm2/met1* mutants (**h**). Scale bars, 50 μm. **i–l** *CLV3* expression patterns in wild-type (**i**), *ago4* (**j**), *ago4/ago6/ago9* (**k**) and *drm1/drm2/met1* mutants (**l**). Scale bars, 50 μm. **m, n** Quantification of SAM size (**m**, *n* = 12 independent plants) and the size of stem cell region (**n**, *n* = 10 independent plants) in wild-type, *ago4*, *ago4/ago6/ago9* and *drm1/drm2/met1* mutants. The error bars indicate the highest and lowest values, the box indicates the middle 50%, the center line indicates the median, the whiskers indicate the data range within 1.5× the interquartile range, and outliers are not shown. Two-sided Student's *t*-tests, *$p < 0.05$, ***$p < 0.001$; ns no significant difference. **o** RT-qPCR quantification of *WUS* and *CLV3* expression levels in wild-type, *ago4*, *ago4/ago6/ago9* and *drm1/drm2/met1* mutants shown as mean ± s.d.; *n* = 3 biological replicates, two-sided Student's *t*-tests, *$P < 0.05$, **$P < 0.01$; ns no significant difference. All experiments were repeated independently at least two times on pools of apices of at least 30 plants, with similar results.

mutants, the size of the *CLV3* domain was more or less unchanged (Fig. 4m, n). Taken together, these results demonstrated that the *AGO4*-mediated RdDM pathway is required for *WUS* function and for maintaining a proper balance between CZ and PZ.

To explore the mechanisms underlying the functional connection of WUS and AGO4, we compared their chromatin binding profiles[38,52] and found that 31% of the AGO4 targets were shared by WUS (255 out of 820 genes) (Fig. 5a and Supplementary Data 4). Given the overlap in expression, function and chromatin binding, we speculated that AGO4 protein may interact with WUS protein in vivo. To test this hypothesis, we carried out Förster resonance energy transfer (FRET) assays using AGO4-mCherry and WUS-GFP, transiently expressed in N. benthamiana leaves. Indeed, we observed a robust interaction of AGO4 and WUS, which we could validate by co-IP (Fig. 5b–d). Interestingly, the WUS homeodomain was sufficient to mediate the interaction with AGO4, suggesting that AGO4 would not compete for binding with other known WUS co-factors, such as HAM1, which has been shown to

bind to the C-terminus of WUS[53]. Furthermore, we found that the AGO4-WUS interaction was disrupted by SNP treatment (Fig. 5c, d), suggesting that NO signaling could directly modify the functional output of WUS.

We next asked whether the interaction of AGO4 and WUS plays a role in SAM control. To this end, we crossed *ago4* into a weak *wus-7* allele, to test for genetic interaction. Supporting the notion that the protein-protein interaction of WUS and AGO4 is important for proper SAM function, we observed a significant reduction in SAM size only in *ago4/wus-7* double mutants, whereas the *CLV3* domain was not changed similar to the *ago* or RdDM triple mutants (Fig. 5e–k). To test the relevance of the RdDM pathway for *WUS* function directly, we turned to *WUS* gain-of-function experiments using the DNA methylation triple mutants as backgrounds. Whereas expressing *WUS* from the *CLV3* promoter in WT plants creates a positive feedback loop leading to the formation of a massively expanded stem cell pool[54], this phenotype was largely suppressed in *drm1/drm2/met1* triple mutants.

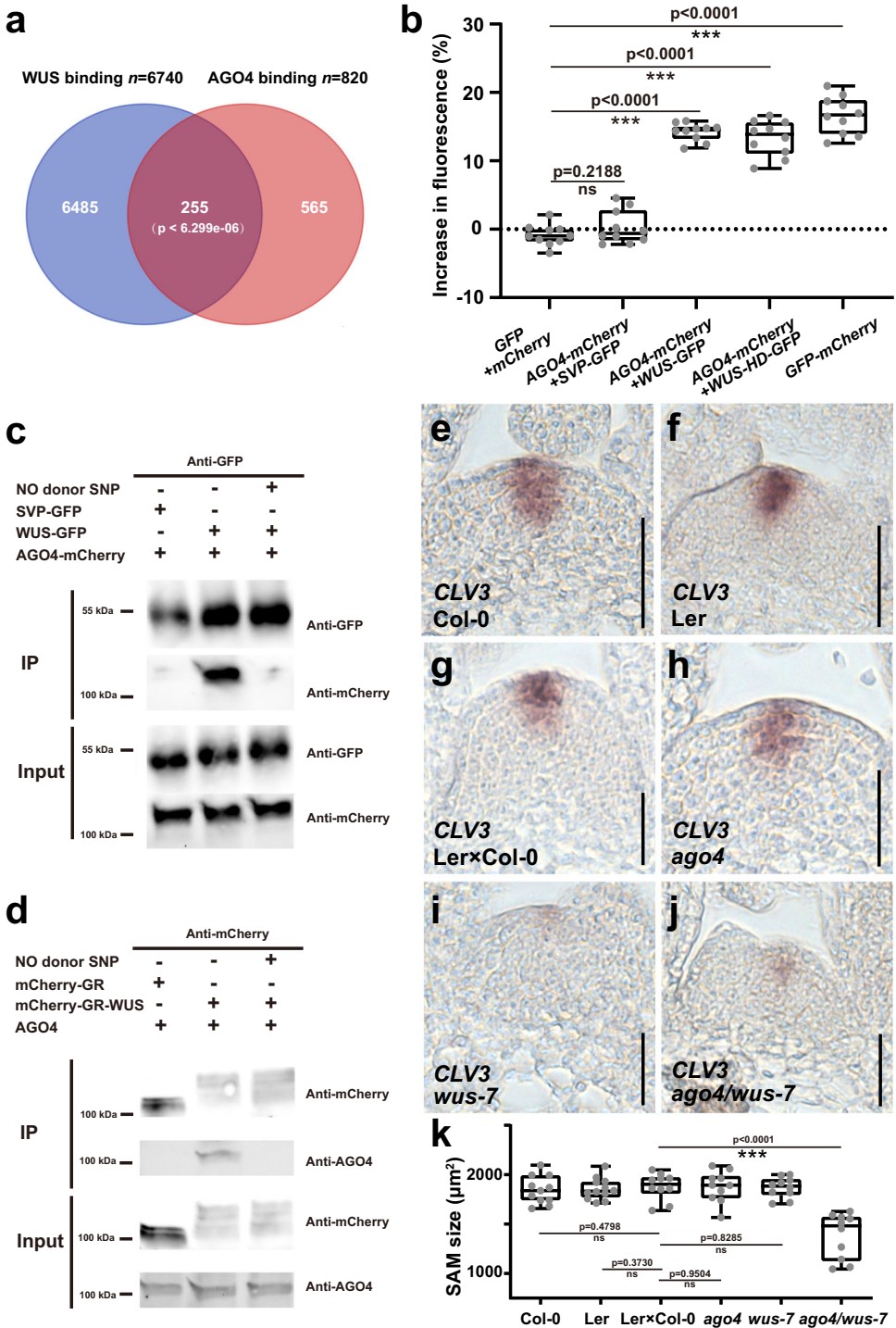

**Fig. 5 | Nitric oxide disrupts the interaction between WUS and AGO4. a** Venn diagram showing the overlap of WUS targets (Ma et al.[38]) and AGO4 targets (Zheng et al.[52]). **b** Analysis of the WUS-AGO4 protein interaction by in planta FRET. The error bars indicate the highest and lowest values, the box indicates the middle 50%, the center line indicates the median, the whiskers indicate the data range within 1.5×the interquartile range, and outliers are not shown. $n = 10$ independent samples, ***$p < 0.001$; ns no significant difference. **c, d** The WUS-AGO4 interaction is disrupted by NO. *p35S::SVP-GFP*, *p35S::WUS-GFP* and *p35S::AGO4-mCherry* were co-transformed into Tobacco leaves (**c**). The expressed proteins were immunoprecipitated using an anti-GFP antibody and then detected with anti-GFP and anti-mCherry antibodies (**c**). Total proteins were extracted from *pUBQ10::mCherry-GR-*

*WUS* and *pUBQ10::mCherry-GR* seedlings upon Dex induction and were immuno-precipitated using an anti-mCherry antibody and then detected with anti-mCherry and anti-AGO4 antibodies (**d**). **e–j** WUS and AGO4 genetically interact. *CLV3* expression patterns in Col-0 (**e**), Ler (**f**), Ler x Col control plants (**g**), *ago4* (**h**), *wus-7* (**i**), and *ago4/wus-7* (**j**) mutants. Scale bars, 50 μm. **k** Quantification of the SAM size. The error bars indicate the highest and lowest values, the box indicates the middle 50%, the center line indicates the median, the whiskers indicate the data range within 1.5× the interquartile range, and outliers are not shown. $n = 10$ independent plants, two-sided Student's *t*-tests, ***$p < 0.001$; ns no significant difference. The experiments in **b–d** were repeated three times, and the experiments in **e–j** were repeated two times on pools of apices of at least 10 plants, with similar results.

Interestingly, in the rare cases we did observe stem cell expansion in this background, we were unable to detect the ectopic expression of *CLV3*, which is a hallmark of this phenotype in WT plants, and the functional readout for ectopic *WUS* activity (Supplementary Fig. 13e, f, k, and l). Consistent with this observation, expression of *WUS* from *CLV3* promoter in *drm1/drm2* significantly suppressed *CLV3* expression (Supplementary Fig. 13e–h), although the phenotypic frequency was comparable to wild-type (Supplementary Fig. 13m). Interestingly, we found WUS to bind to many transposable elements with roughly 4000 binding events genome wide (Supplementary Fig. 14), suggesting that WUS could participate in repressing the activity of these elements. Taken together, these results demonstrated that AGO4 physically interacts with WUS and that this interaction, as well as the DNA methylation machinery of the RdDM pathway, is required for WUS activity and proper SAM function.

### Nitric oxide controls expression of key meristem regulators via DNA methylation

Given the fact that AGO4 is able to direct DNA de novo methylation in plants[45–47] and having shown that NO controls AGO4 at the transcriptional and post-translational levels, we speculated that some of the effects of NO we had observed were due to changes in DNA methylation patterns. To test this hypothesis, we treated *ap1/cal* meristems with SNP and quantified global DNA methylation. Indeed, DNA methylation levels were significantly reduced after SNP treatment (Fig. 6a), in line with the regulatory interaction of NO with AGO4. Since we found AGO4 and WUS to interact, we next tested whether DNA methylation at shared targets would be affected by NO signaling. To this end, we carried out whole-genome bisulfite sequencing on meristems that were treated with SNP or mock and found a total of 16,301 differential methylated regions (DMRs) (Supplementary Data 5). Among them, 96% of DMRs were in the CHH context, which is mediated by the RdDM pathway (Supplementary Fig. 15), indicating that NO affects RdDM activity in the SAM. Zooming in on the co-targets of WUS and AGO4, we found that 168 of 255 (65%) showed a significant reduction in DNA methylation (Fig. 6b and Supplementary Data 4). These results suggested that NO could affect meristem function by blocking de novo DNA methylation at WUS targets via the inhibition of AGO4. In this scenario, the expression of WUS targets relevant for SAM function should be enhanced following activation of NO signaling in the SAM, or perturbation of the RdDM pathway. To test this directly, we examined the 26 NO-induced plant stem cell-related genes identified here and previously[25] in our methylation data and identified nine genes (*ARR7, CLE13, CLV2, ERL2, ERL1, AT3G15150, BARD1, MEE22* and *CRN*), which exhibited substantially decreased DNA methylation following SNP treatment (Supplementary Fig. 16).

Since we had shown previously that *ARR7* is directly repressed by WUS and has important roles for SAM function[35,55,56], we decided to test whether *ARR7* is also downstream of NO signaling and the RdDM pathway. To this end, we examined *ARR7* expression in SAMs of wild-type, *ago4, ago4/ago6/ago9, drm1/drm2, rdm1* and *drm1/drm2/met1* mutants with and without SNP treatment. We observed a significant and consistent increase of *ARR7* mRNA accumulation after SNP treatment in wild-type (Fig. 6d, h, l and Supplementary Figs. 17–19), which confirmed RNA-seq results and fitted well with the finding that SNP treatment reduced DNA methylation at the *ARR7* promoter (Fig. 6c, d). Consistent with an important role for *AGO* genes and the RdDM pathway in repressing *ARR7* expression, *ARR7* mRNA also accumulated in the respective mutants without SNP treatment. Indeed, quantification of *ARR7* mRNA levels by RT-qPCR demonstrated that the SNP effect on *ARR7* expression was reduced in *drm1/drm2* and *rdm1* mutants (Supplementary Fig. 19), while it was fully repressed in *ago* and DNA methylation triple mutants, demonstrating that the effect of NO is largely mediated by the RdDM pathway (Fig. 6m). As we showed

that AGO4 interacts with WUS in vivo, we wondered if AGO4 is recruited to the same sites of *ARR7* promoter as WUS. To test this idea, we carried out ChIP experiments using mCherry-AGO4 rescue plants, and observed that binding of AGO4 to the *ARR7* promoter overlaps with one of the regions also bound by WUS (Supplementary Fig. 20).

To further test the role of RdDM pathway in transmitting NO signals in the SAM, we next selected stem cell- and niche cell specific genes and examined their expression in the wild-type and in *ago4, ago4/ago6/ago9, drm1/drm2, rdm1* and *drm1/drm2/met1* mutants under SNP treatments. To our surprise, of the 12 stem cell- and niche-specific NO responsive genes, about half lost their response to SNP treatment in the *drm1/drm2, rdm1,* and *ago4/ago6/ago9* mutants. Similarly, nine completely lost their response in the *drm1/drm2/met1* mutants (Supplementary Fig. 21). Taken together, our results suggested that RdDM pathway downstream of AGO4 mediates NO signaling in the SAM and is essential for proper expression of core stem cell regulators.

## Discussion

Nitric oxide has been shown to act as an important signaling molecule in highly diverse contexts, including priming cell fate, or defense across the plant and animal kingdoms[14–20]. Growing evidence suggests that NO is involved in stem cell regulation in animals[15,27,28], although the molecular mechanisms so far remain largely elusive. Our results now reveal a central role for NO in the communication between the transit amplifying cells at the periphery and the stem cells in the center of the shoot stem cell system of *Arabidopsis*. Remarkably, NO signaling appears to restrict expression and activity of AGO4, a key component of the small RNA-directed DNA methylation pathway, to the center of the meristem. The RdDM pathway is an evolutionary conserved epigenetic machinery for gene silencing and genome stability, and is involved in diverse developmental processes in both plant and animals[57–65]. Recent evidence suggests that DNA methylation plays an important role for shoot and root meristem function and is likely more dynamic than anticipated[66–69]. We had previously shown that stem cell fate is dependent on large scale repression of gene expression by WUS, which could partially be explained by histone de-acetylation[35,36,38]. Our finding that WUS physically interacts with AGO4 and that this interaction is interrupted by NO now suggests that part of the repressive activity of WUS may be mediated by DNA methylation. We hypothesize that AGO4 and in turn the DNA methylation machinery is directed to WUS target loci via this interaction. One prominent example is *ARR7*, a well-studied direct negative WUS target, whose expression is substantially elevated in mutants of either *AGO* or *DNA* methylation genes. Importantly, the expression of the core stem cell signaling component *CLV3* is enhanced by WUS[4–9] and consistently, is not directly affected by NO signaling or mutations in the RdDM pathway. Taken together, the identification of the NO WUS-AGO4 interaction limited to the center of the meristem by NO may be able to explain the divergent functions of WUS in cells of the center versus the periphery. Although the first molecular link between NO signaling and stem cell fate in plants has now been established, many open questions remain. One pressing example is the regulation of WUS expression by NO and in this context, it will be important to elucidate the interaction of NO with the CLE40/BAM1 signaling system[13]. Furthermore, the idea that AGO4 can be recruited to chromatin by interaction with a transcription factor in addition to the canonical mechanism involving small RNAs, suggests that the RdDM pathway has been hijacked for developmental processes. Another striking facet is the realization that WUS binds to chromatin regions containing transposable elements, raising the possibility that this activity provides an additional layer or protection against mobilization of mobile genetic elements in stem cells. Initial studies using AlphaFold[70,71] to model the AGO4-WUS interaction suggest that in addition to an interaction of the WUS homeodomain with

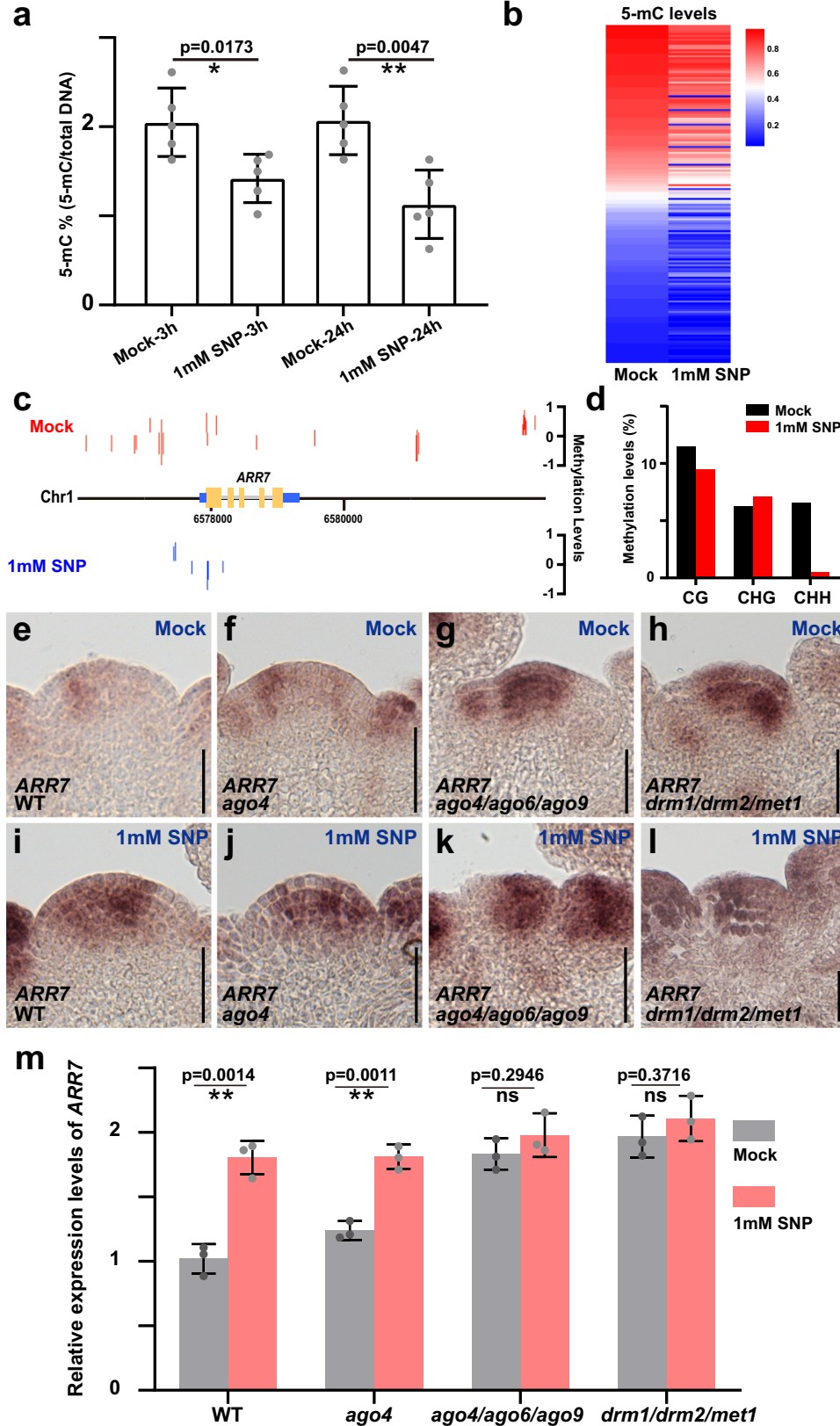

AGO4, the largely unstructured part C-terminal of the homeodomain may be inserted into the central cleft of AGO4, which usually interacts with small RNAs. These findings now provide ample opportunity for detailed molecular studies, and open fascinating avenues to study the dynamics of epigenetic modifications.

## Methods

### Plant materials and growth conditions

*Arabidopsis thaliana* ecotype Columbia-0 (Col-0) was used as the wild-type excepted where noted. The *mCherry-AGO4* reporter was kindly provided by P. Jullien; the *nia1/nia2/noa1* seeds by J. León; the *ago4-5*,

**Fig. 6 | RdDM activity mediates NO signaling via DNA methylation.**
**a.** Quantification of global 5-mC DNA methylation levels of wild-type plants after mock and SNP treatments using ELISA with five replicates ($n = 10$ independent shoot apices for each replicate). The data are shown as mean ± s.d.; two-sided Student's $t$-tests, *$p < 0.05$, **$p < 0.01$. **b** Heatmap of 5-mC levels in the promoter region of shared WUS and AGO4 targets after mock and SNP treatments showing the trend towards reduced methylation. **c** Annotation of 5-mC sites in the *ARR7* genomic region after mock and SNP treatment demonstrating the loss of DNA methylation. **d** Quantification of CG, CHG, and CHH methylation levels at *ARR7*.

**e−h** *ARR7* expression patterns in wild-type (**e**), *ago4* (**f**), *ago4/ago6/ago9* (**g**) and *drm1/drm2/met1* (**h**) after mock treatment. Scale bars, 50 μm. **i−l** *ARR7* expression patterns in wild-type (**i**), *ago4* (**j**), *ago4/ago6/ago9* (**k**) and *drm1/drm2/met1* (**l**) after SNP treatments. Scale bars, 50 μm. **m** Quantification of *ARR7* mRNA levels in wild-type, *ago4*, *ago4/ago6/ago9* and *drm1/drm2/met1* after mock and SNP treatments shown as mean ± s.d.; $n = 3$ biological replicates, two-sided Student's $t$-tests, **$P < 0.01$; ns no significant difference. All experiments were repeated independently at least two times on pools of apices of at least 30 plants, with similar results.

*ago4-6/ago6-2/ago9-1* seeds by M. Axtell; the *rdm1* seeds by H. Ito; the CLV3 and WUS reporters were generated as described before[72]. All seeds were sterilized by applying 70% ethanol and 0.5% Tween for 10 mins, followed by washing two times with 95% ethanol and air drying. Afterwards, the seeds were sown on 0.8% Phyto agar plates (half-strength Murashige and Skoog (MS) salts and 1% sucrose), kept at 4 °C two overnights for imbibition. After germination, plants were transferred to soil and grown under long-day conditions (16 h day/8 h night regime at 21 °C).

## Cloning
All constructs were generated using GreenGate cloning system[73]. Details about the cloning of the modules and assembled plasmids are listed in Supplementary Data 6.

## In situ Hybridization
Templates for RNA probes were amplified from cDNAs using gene-specific primers containing T7 or T3 promoter sequences at the 5′ end. A complete list of primer sequences can be found in Supplementary Data 7. The RNA probes were synthesized by T7 RNA polymerase, and in situ hybridization was performed in accordance with standard protocols[74].

## RNA sequencing and analysis
Shoot apices from 6-week inflorescences of *ap1/cal* (Col-0) with and without SNP treatments were collected. Total RNA (10 μg) was subjected to DNase I treatment, and mRNA was kit purified (Life Technology). Strand-specific RNA-seq libraries were prepared using the NEBNext® Ultra™ RNA Library Prep Kit for Illumina following the manufacturer's instructions and were subjected to NextSeq550 for 75 bp end sequencing. Reads were aligned to the *Arabidopsis* TAIR10 reference genome using STAR, and differentially expressed genes were called using DEGseq[75] with the criterion of fold-change ≥1.5 and $p$-value < 0.01. For RNA sequencing, each condition was analyzed in three biological replicates. The raw data have been deposited in the NCBI Gene Expression Omnibus (GEO) under the accession numbers GSE216952 and GSE243447.

## Enzymatic methyl-seq (EM-seq) sequencing and analysis
Genomic DNA from *ap1/cal* shoot apices with and without 24 h SNP treatments, was extracted using a DNeasy Plant Mini Kit (Qiagen), and was sheared to approximately 250 bp fragments. EM-seq libraries were prepared from sheared DNA using an enzymatic methyl-seq kit following the standard instructions (New England BioLabs), and were subjected to NextSeq550 using 75 bp paired-end sequencing. The C to T conversion rates were 99% and the genome coverage was 10× for the EM-seq data. Methylated cytosines were extracted from aligned reads using the Bismark methylation extractor with default parameters. All analyses were done using sites covered by a minimum of 10 reads in both samples. Only 5-mC sites supported by two biological replicates were considered for further analysis. The DNA methylation levels were calculated as described before[76]. Briefly, DNA methylation level at each site or region was calculated by number of C vs total C and T account. The raw data have been deposited in the NCBI Gene Expression Omnibus (GEO) under the accession number GSE216952.

## Chromatin immunoprecipitation (ChIP)
Chromatin immunoprecipitation (ChIP) was performed on *pUB-Q10::mCherry-GR-WUS* and mCherry-AGO4 rescue plants. ChIP was conducted as previously described with minor modifications[77]. A Diagenode Bioruptor UCD-200 was used for sonication (30 s on, 30 s off, medium, 15 min duration; sonication buffer: 10 mM $Na_3PO_4$, 100 mM NaCl, 0.5% sarkosyl, 10 mM EDTA, 1 mM PMSF, 1 tablet per 10 ml, pH 7). GR antibodies (Santa Cruz, sc-393232) and RFP-Trap Magnetic Agarose (ChromoTek, rtma-20) were used to precipitate chromatin, and no antibody was used as negative controls.

## Electrophoretic mobility shift assay (EMSA)
The homeodomain of *WUS* was cloned behind a 6xHis-MBP tag. The protein was produced in *Escherichia coli* strain Rosetta and purified with nickel Sepharose media (GE Healthcare). EMSA was performed according to the reference guide of the LightShift Chemiluminescent EMSA kit (Thermo Scientific 20148), and probes were labeled with biotin. The gene-specific primer sequences are listed in Supplementary Data 7.

## FRET
FRET was measured by using increase in donor fluorescence upon bleaching of the acceptor as readout. Three independent experiments were performed, and per experiment, at least 20 cells per sample were analyzed. To drive expression in *N. benthamiana* leaves, the *35S* or *UBQ10* promoter was used.

## Confocal microscopy
For imaging of *pCLV3::mTagBFP2, pWUS::2xVenus-NLS, pUBQ10::mCherry,* and *pH3.3::GFP*, inflorescences were dissected and imaged on an upright Nikon A1 Confocal with a CFI Apo LWD 25 × water immersion objective (Nikon Instruments) without coverslip as described[38].

## Global DNA methylation assay
The global DNA methylation levels were measured by using a Global DNA Methylation Assay Kit (abcam, ab233486) following the manufacturer's protocol.

## Accession numbers
The Arabidopsis Genome Initiative numbers for the genes mentioned in this article are as follows: *NIA1* (AT1G77760); *NIA2* (AT1G37130); *NOA1* (AT3G47450); *WUS* (AT2G17950); *CLV3* (AT2G27250); *AGO4* (AT2G27040); *AGO6* (AT2G32940); *AGO9* (AT5G21150); *DRM1* (AT5G15380); *DRM2* (AT5G14620); *MET1* (AT5G49160); *ARR7* (AT1G19050); *RDM1* (AT3G22680); *UFO* (AT1G30950); *CUC2* (AT5G53950); *AP1* (AT1G69120); *CAL* (AT1G26310).

## Statistical analysis
Differences between groups were identified using two-sided Student's $t$-test, and $p$-values less than 0.05 were considered statistically significant. Details and sample sizes as well as the numbers of biological replicates are provided in the respective figure legends.

## Reporting summary
Further information on research design is available in the Nature Portfolio Reporting Summary linked to this article.

## Data availability

The data underlying Figs. 1m, n; 2m; 3p; 4m, n; 5b–d, k; 6a and Supplementary Figs. 1g; 3d; 5d, h; 6h, i; 8e; 12j, k; 13m are provided in the Source Data file. RNA-seq, EM-seq, and ChIP-seq raw data have been deposited in the NCBI Gene Expression Omnibus (GEO) under the accession numbers GSE216952 and GSE243447. Source data are provided with this paper.

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

## Acknowledgements

We thank P. Jullien for sharing the mCherry-AGO4 transgenic line before publication, J. León for sharing the *nia1/nia2/noa1* seeds, M. Axtell for sharing the *ago4-5* and *ago4/ago6/ago9* seeds, H. Ito for sharing the *rdm1* seeds. We thank Z. Zhao for stimulating discussion and advice. This work was supported by the ERC through the Synergy Grant 810296 "DECODE" to J.U.L.

## Author contributions

J.Z., X.Z., and J.U.L. designed the experiments, analyzed the data, and wrote the paper with input from all other authors. X.Z. performed the RNA-seq, the BS-seq, and the EMSA experiments. Z.L. analyzed the RNA-seq and the BS-seq data. I.H., M.G., and S.G. preformed and analyzed ChIP-seq. P.F. conducted the FRET assays. C.W. generated the reporter lines for WUS and CLV3. S.G. performed bioinformatic analyses. J.Z. conducted all the other experiments.

## Funding

## Competing interests

The authors declare no competing interests.
