## [Peer Review File · Nature Communications]

Nitric Oxide controls shoot meristem activity via regulation of DNA methylationREVIEWER COMMENTS

Reviewer #1 (Remarks to the Author):

The data presented in this manuscript by Zeng et al. have focused on roles of nitric oxide (NO) in regulation of shoot meristem activity in a manner of DNA methylation in Arabidopsis. Several lines of genetic evidence reveal that NO signaling controls plant stem cell homeostasis using a set of mutant materials. Importantly, genetic, biochemical and molecular tests suggest that the stem cell inducing WUSCHEL transcription factor directly interacts with AGO4 in an NO-dependent manner, which has provided molecular linkage to connect two signaling systems to modify DNA methylation patterns. Notably, in combination with phenotypic analysis of specific mutants, detecting specific and dynamic changes in NO generation around shoot meristem region have provided insights into better understanding the role of NO in represses central cell fate, likely via repressing WUS expression. This study appears as a promising starting point to supply lines of strong evidence to recognize the requirement of the regulation of WUS expression by NO in shoot meristem cell fate decisions.

Major comments:

(1) “we continuously increased NO levels in the CZ by expressing NOA1, one of the NO biosynthesis gene, from the CLV3 promoter (Supplementary Fig. 4a-e)”. Actually, no images or quantification data of DAF-2DA fluorescent intensity were shown in Supplementary Fig. 4 to specify the increase in NO levels in the indicated CLV3::NOA1 transgenic plants compared with WT.

(2) To test this directly, we examined the 26 NO-induced plant stem cell-related genes identified here and previously²⁵ in our methylation data and identified nine genes (Supplementary Fig. 9”).

Are there any common responsive elements located in the promoters of these 26 NO-induced plant stem cell-related genes?

(3) “We further tested the transcriptional response of NIA1, NIA2 and NOA1 to ectopically induced WUS activity and found that at the whole seedling level NIA1 and NOA1 showed a robust repression after 4 hours WUS of induction (Supplementary Fig. 3a-c)”.

Have the authors tested the binding activity of WUS to the promoter of NOA1 by ChIP-PCR or EMSAs?

(4) “we took advantage of the ap1/cal mutant, which is characterized by a massive over-proliferation and subsequent arrest of inflorescence and young floral meristems”.

The authors performed the RNA-seq analysis on microdissected apices with and without SNP treatment for 24 hours with the ap1/cal mutant. The question is that how many typical NO-responsive genes identified in whole seedlings can be repeatedly detected in the ap1/cal mutant. That is the key point to confirm whether the transcriptome profiling performed with the ap1/cal mutant is acceptable in relative to the WT whole plants.

Minor comments:

(1) Page 10, line 10, 13, the AGO4 could be modified by S-nitrosylation (?) in response to NO.

(2) What is the concentration of cPTIO used in experiments?

(3) In Fig. 1 | Nitric oxide is required for proper meristem activity. Quantification of SAM size (m). What is the length unit for SAM size?

Reviewer #2 (Remarks to the Author):

Zeng et al. present a very intriguing finding about the connection of NO signaling to stem cell function and provide a mechanistic model involving the canonical stem cell maintenance factor WUS and the small RNA effector AGO4 and its RdDM activity. This work could be of broad interest to researchers interested in stem cell biology and epigenetics from the animal and plant fields.

The Authors show that the central zone of the SAM contains less NO, likely due to less activity of NO biosynthetic genes NIA1, NIA2, and NOA. Reduced NO levels result in smaller meristems with more WUS and CLV3 expression. They also used a pharmacological approach to show that NO levels can modify WUS expression. Next, they show that WUS directly represses NIA1. Using transcriptome data, they identify AGO4 as a probable target of NO signaling. Therefore they investigate the meristem of mutant plants but cannot see differences in *ago4*, but only in *ago4ago6ago9* or *drm1drm2met1* triple mutants. Next, they find that AGO4 and WUS can interact and, given the involvement of AGO4 in RdDM, investigate DNA methylation in SNP-treated meristematic material. They find a reduction of DNA methylation and conclude that SNP interrupts the AGO4-WUS repressive activity.

Major concerns:

The interaction of AGO4 with WUS and the possibility that AGO4 is recruited with or without loaded small RNAs by WUS to the promoter of genes is fascinating. But the interaction assays rely on transiently overexpressed proteins. Therefore, this interaction should be confirmed in *Arabidopsis* with, e.g., *pAGO4:AGO4cherry* and *pWUS:WUS-GFP* or at least with their inducible system for WUS. AlphaFold could also add additional evidence for an interaction.

They should also investigate if this proposed chromatin tethering of AGO4 by WUS would depend on small RNAs (e.g., using *poliv*) or not. To show that AGO4 is recruited to the same sites as WUS (e.g., to promoters of NIA1 or ARR7) by chip-pcr would be affirmative. Also, the promoters of potential target genes should be analyzed if they carry transposon fragments as target sites for RdDM.

They should also investigate the existence of small RNAs at the promoter regions of common AGO4/WUS targets.

Less importantly, it would also be interesting to investigate a potential connection to 22nt small RNAs, as there is a strong association between the silencing of NIA1 and NIA2 and the activity of small RNAs (Wu et al., *Nature* 581, 2020).

Another concern is the analysis and interpretation of the DNA methylation data. First, AGO4 is almost entirely responsible for RdDM-mediated CHH methylation (Stroud et al., 2013).

Still, many effects the authors claim to be connected to RdDM are visible only in *ago4/ago6/ago9*. Also, the choice of *drm1/drm2/met1* is unfortunate, as *met1* has global DNA methylation defects, not limited to RdDM. A better option would have been strict RdDM mutants e.g., *drm1/drm2*, *rdr2*, *rdm1*, or *nrpe1*. Could it be that the observed transcriptional changes are not related to RdDM but to additional functions of AGO6 and AGO9 or MET1? The authors should at least use one more RdDM mutant to reduce this concern.

It would also be excellent to see potential changes in promoter CHH methylation of NIA1 and NIA2 plus SNP.

Minor concerns:

Paragraph "Nitric oxide promotes peripheral zone fate via repression of WUS"

It is hard to see in Supplementary Fig. 1a the claim that NO accumulated to higher levels in the PZ compared to the CZ. Please indicate fluorescence intensities across the picture and show more examples of meristems.

Supplementary Fig. 2: the Signal for NIA2 is challenging to see. Given the difficulties of producing clear in situ, please provide at least antisense and/or knock-out mutant controls for all three genes.

To evaluate the functionality of all three genes: Did the authors observe phenotypic differences to wt in any of the double mutant combinations? For example, do *nia1/nia2/noa1* plants grow slower, or are they also producing fewer organs (do they flower with the same or a different number of leaves?)? Are they growing normally on soil, or need additional nitrogen sources?

It looks like the experiment in Supplementary Figure 1a-d has been done only with one replica. Either clarify or provide more biological replicas. It would be beneficial to show this experiment performed with 35S::ALCR;ALA::GUS to exclude the influence of ethanol on the staining.

Fig1 m,n): On which data is the quantification based? Is it linear measurements of in-situ hybridizations? Please clarify.

Supplementary Fig.5 comes right after supplementary Fig.2. Please make the order chronological to make reading easier.

More importantly, Supplementary Fig.5 is an important control experiment to exclude a non-specific effect of NO on WUS expression. But it is not possible to judge from the images the validity. Fluorescence intensities from several meristems need to be quantified.

The line "We inspected our resources on direct WUS target genes". Which resources?

RNA-seq data: please describe briefly in the text how DEGs are defined (which statistical test was used, what is the TPM/padj/foldchange cutoff?)

Fig.3: a: it is confusing to have both SNP/Mock down and SNP/Mock up in the same Venn diagram. Either show only the down or split them, or use upset plots.

AGO4 is one of 10 AGO proteins most highly expressed in the SAM. Therefore it would be essential to show the specificity of their probe by using knock-out *ago4* plants.

Figure 3o: please indicate the SNP concentration.

The authors report that AGO4 accumulates in the cytoplasm upon SNP treatment. Are the S-nitrosylation sites on a potential NLS or NES?

Chapter "AGO4 protein connects WUS function with the RdDM pathway"

Sentence: "together with its orthologues AGO6 and AGO9. The functional relevance of AGO9 for RdDM has never been demonstrated.

As mentioned already above, the usage of *drm1/drm2/met1* was not a good choice. MET1 is a CG maintenance methyltransferase, and loss of MET1 has all kinds of secondary effects (e.g., late flowering). It would have been better to use other RdDM mutants like *drm1/drm1*, *rdr2*, or *poliv*.

Figure 5a: What is the p-Value of the overlap (e.g., use a hypergeometric test). The reference for Zheng et al. should be in the text and the reference list.

Figure 5c: Please indicate the antibody used for IP in the Figure.

The IP should be repeated using Arabidopsis stably transformed Arabidopsis plants.

Which *ago4* allele was used? Supplementary Fig. 7: please write which mutant is in which genetic background. It would also be important to show the SAM size of Ler and Col wt plants. I assume *ago4wus7* plants are from an F2 population from the *ago4* x *wus7* cross. Are the LerXCcol plants also F2? LerXCcol f2 plants are problematic as they are very

heterogeneous. The better experiment would have been to generate a CRIPR ago4 knock-out in the wus7 background.

Line " this phenotype was largely suppressed in drm1/drm2/met1 triple mutant": please refer to picture. Is it Supplementary Fig. 8 c/d?

Unfortunately, Supplementary Fig. 8 is also difficult to interpret as Met1 is not involved in RdDM but DRM1/DRM2, meaning the conclusion that "the RdDM pathway is required for WUS activity and proper SAM function" is not supported by the data. They could repeat this experiment with other RdDM mutants (as suggested above). It is also unclear to me how the authors interpret Supplementary Fig.8: Does that mean DNA methylation is required to activate CLV3? That would be in contrast to the fact that most DNA methylation mutants develop similarly to wt.

Chapter: "Nitric oxide controls expression of key meristem regulators via DNA methylation"

The DNA methylation data need to be analyzed more carefully. Which treatment length of SNP was used for Bisulfite sequencing? It is critical to distinguish between CG, CHG, and CHH methylation, as only CHH methylation is affected by the loss of AGO4. What kind of DNA methylation levels do the authors describe? Weighted DNA methylation? It is unclear to me how Fig. 6 b shows a trend towards "reduced DNA methylation"?

Supplementary Fig.9: I need clarification on why the authors show total DNA methylation levels, as this is irrelevant to AGO4 function. They should distinguish CG, CHG, CHH methylation, and promoter and gene body methylation. It is also unclear to me how "total 5-mC" was calculated, as it is unclear from the description in M&M. Please use methylation levels as defined in Schultz et al., Trends Genet 2012., or describe better how the calculation was done.

To ease the reproducibility of the study, please include the atg numbers plus Stock-identifiers for all genes and mutants used in the study.

Discussion:

I do not know about RdDM activity in animals; the papers they refer to do not show this.

Methods: Please provide the C to T conversion rates (using the chloroplast genome) and genome coverage for the EM-seq data.

Reviewer #3 (Remarks to the Author):

The manuscript by Zang et al. entitled "Nitric Oxide controls shoot meristem activity via regulation of DNA methylation" reports the roles of NO and RdRM in the SAM. AGO4 is inhibited by NO at the transcriptional level and also post-transcriptional level, which is S-nitrosylation dependent. AGO4 interacts with WUS at the protein level, and this interaction is disrupted by NO treatment. The roles of RdRM in the SAM have been reported. For example, rice RdRM mutant plants have prominent SAM defects. This study links NO regulation to the roles of RdRM and WUS. Together, this study provides many interesting clues but there seems to be insufficient support for the models proposed by the authors. In the model, many links are not well established either. Below are my specific comments.

1. How SAM size, in particular PZ size was determined? UFO has been used as a PZ marker (Lee et al., 1997 Curr Biol doi: 10.1016/s0960-9822(06)00053-4). A recent work shows AS2 expression in the PZ as well (Burian et al., 2022 Nat Plants doi: 10.1038/s41477-

022-01111-3). Or KAN1 highlighting the outer boundary of the SAM.

2. Fig. 2a-c, 3 h treatment w/ SNP or cPTIO causes WUS signal changes. Was the pWUS::2xVenusNLS reporter used? If so, the reduction after SNP treatment seems suggest a fast degradation of 2xVenusNLS. Why Venus is so unstable? Note that the authors have shown that GFP fluorescence is not responsive to NO levels (Suppl Fig 5a-c). Venus is a fast maturing protein, but 3 h is still short for a substantial portion of Venus to mature (Balleza et al, 2018 Nat Methods doi: 10.1038/nmeth.4509).

3. The protein interaction between WUS and AGO4 is interesting. However, the biological relevance is less clear. If the interaction contributes to WUS activity as a transcription factor, differential expression of WUS-target genes should be identified in ago4 mutant plants. Alternatively, if WUS participates in the RdRM activity of AGO4, AGO4 targets should be different in wus mutants or WUS-OX lines.

4. The DAF-2DA staining results is not very clear. Is the signal enriched in incipient primordia or the boundaries between primordia?

5. The expression of both WUS and CLV3 are enhanced in the nia triple mutant. Why?

6. In Fig. 2j-l, CLV3 signal seems to be enhanced after SNP and cPTIO treatment.

7. To really test whether S-nitrosylation is responsible for AGO4-WUS interaction, the authors need to disrupt all possible S-nitrosylation sites, which can be identified by mass spectrum analysis.

Point-by-point responses to the reviewer comments

We would like to thank all reviewers for their insightful comments, which were very helpful to improve the manuscript. Following their advice, we have carried out a large number of new experiments and have extensively revised the manuscript. Specific responses to all comments can be found below:

-- Reviewer #1:

The data presented in this manuscript by Zeng et al. have focused on roles of nitric oxide (NO) in regulation of shoot meristem activity in a manner of DNA methylation in Arabidopsis. Several lines of genetic evidence reveal that NO signaling controls plant stem cell homeostasis using a set of mutant materials. Importantly, genetic, biochemical and molecular tests suggest that the stem cell inducing WUSCHEL transcription factor directly interacts with AGO4 in an NO-dependent manner, which has provided molecular linkage to connect two signaling systems to modify DNA methylation patterns. Notably, in combination with phenotypic analysis of specific mutants, detecting specific and dynamic changes in NO generation around shoot meristem region have provided insights into better understanding the role of NO in represses central cell fate, likely via repressing WUS expression. This study appears as a promising starting point to supply lines of strong evidence to recognize the requirement of the regulation of WUS expression by NO in shoot meristem cell fate decisions.

RESPONSE: We thank the reviewer for pointing out the importance of our work.

Major comments:

(1) “we continuously increased NO levels in the CZ by expressing NOA1, one of the NO biosynthesis gene, from the CLV3 promoter (Supplementary Fig. 4a-e)” . Actually, no images or quantification data of DAF-2DA fluorescent intensity were shown in Supplementary Fig. 4 to specify the increase in NO levels in the indicated CLV3::NOA1 transgenic plants compared with WT.

RESPONSE: We have now included new data to show the distribution of NO in *CLV3::NOA1* transgenic plants, along with additional pictures of wt including careful quantification of staining levels. We have added this information in the revised Supplementary Fig 6. After DAF-2DA staining, we observed that endogenous nitric oxide level was increased in the stem cell domain in the *CLV3::NOA1* transgenic plants.

(2) To test this directly, we examined the 26 NO-induced plant stem cell-related genes identified here and previously in our methylation data and identified nine genes (Supplementary Fig. 9) .

Are there any common responsive elements located in the promoters of these 26 NO-induced plant stem cell-related genes?

RESPONSE: As suggested, we carried out an analysis of known regulatory elements the 26 NO-induced plant stem cell-related genes and found a number of elements to be represented, including light, stress, and hormone responsive elements. We have added this new analysis to Supplementary Fig. 16b. Due to the small number of genes, a de novo analysis of potential response elements is not possible.

(3) “We further tested the transcriptional response of NIA1, NIA2 and NOA1 to ectopically induced WUS activity and found that at the whole seedling level NIA1 and NOA1 showed a robust repression after 4 hours WUS of induction (Supplementary Fig. 3a-c)” .

Have the authors tested the binding activity of WUS to the promoter of NOA1 by ChIP-PCR or EMSAs?

RESPONSE: We have carried out a new WUS ChIP-seq experiment based on the much improved eChIP protocol. Based on this very much improved data, we find strong evidence that WUS protein binds to the promoters of NIA1 and NIA2, but not NOA1. This data is now shown in the new Supplementary Figure 7.

(4) “we took advantage of the *ap1/cal* mutant, which is characterized by a massive over-proliferation and subsequent arrest of inflorescence and young floral meristems” .

The authors performed the RNA-seq analysis on microdissected apices with and without SNP treatment for 24 hours with the *ap1/cal* mutant. The question is that how many typical NO-responsive genes identified in whole seedlings can be repeatedly detected in the *ap1/cal* mutant. That is the key point to confirm whether the transcriptome profiling performed with the *ap1/cal* mutant is acceptable in relative to the WT whole plants.

RESPONSE: As suggested, we performed the RNA-seq with seedlings of WT (*Col-0*) and *ap1/cal* mutants with and without 24 h SNP treatment to test for comparability of wt and *ap1/cal* mutants with regards to NO response. We observed that WT and *ap1/cal* mutants share 88% of responsive genes after SNP treatment(Supplementary Fig. 9), suggesting that the NO response of WT and *ap1/cal* mutants is comparable. We added the new data to our revised manuscript in the new Supplementary Fig. 9.

Minor Comments:

(1) Page 10, line 10, 13, the AGO4 could be modified by S-nitroslation (?) in response to NO.

RESPONSE: Yes, the AGO4 protein could be modified by S-nitroslation as we observed that its

protein has multiple S-nitrosylation sites by *in vivo* biotin-switch assay (Fig. 3p).

(2) What is the concentration of cPTIO used in experiments?

RESPONSE: We thank the reviewer's point and apologize for omitting this important information. We already added the information in the revised manuscript.

(3) In Fig. 1 | Nitric oxide is required for proper meristem activity. Quantification of SAM size (m). What is the length unit for SAM size?

RESPONSE: Thank you for bringing up this important point. We have now included this information in the revised figures.

--Reviewer #2:

Zeng et al. present a very intriguing finding about the connection of NO signaling to stem cell function and provide a mechanistic model involving the canonical stem cell maintenance factor WUS and the small RNA effector AGO4 and its RdDM activity. This work could be of broad interest to researchers interested in stem cell biology and epigenetics from the animal and plant fields.

The Authors show that the central zone of the SAM contains less NO, likely due to less activity of NO biosynthetic genes NIA1, NIA2, and NOA. Reduced NO levels result in smaller meristems with more WUS and CLV3 expression. They also used a pharmacological approach to show that NO levels can modify WUS expression. Next, they show that WUS directly represses NIA1. Using transcriptome data, they identify AGO4 as a probable target of NO signaling. Therefore they investigate the meristem of mutant plants but cannot see differences in *ago4*, but only in *ago4ago6ago9* or *drm1drm2met1* triple mutants. Next, they find that AGO4 and WUS can interact and, given the involvement of AGO4 in RdDM, investigate DNA methylation in SNP-treated meristematic material. They find a reduction of DNA methylation and conclude that SNP interrupts the AGO4-WUS repressive activity.

RESPONSE: We thank the reviewer for pointing out the broad impact and novelty of our work.

Major concerns:

The interaction of AGO4 with WUS and the possibility that AGO4 is recruited with or without loaded small RNAs by WUS to the promoter of genes is fascinating. But the interaction assays rely on transiently overexpressed proteins. Therefore, this interaction should be confirmed in *Arabidopsis* with, e.g., pAGO4:AGO4cherry and pWUS:WUS-GFP or at least with their inducible system for WUS. AlphaFold could also add additional evidence for an interaction.

RESPONSE: As suggested, we now performed the co-IP experiment in *Arabidopsis* using

UBQ10::mCherry-GR-WUS plants to confirm the interaction of AGO4 and WUS *in vivo*. The results showed that endogenous AGO4 indeed interacts with WUS in seedling (Fig. 5d). In addition, we could also observe the interaction between AGO4 and WUS by AlphaFold prediction, which suggest that the homeodomain, as well as the disordered C-terminus of WUS are in contact with AGO4.

They should also investigate if this proposed chromatin tethering of AGO4 by WUS would depend on small RNAs (e.g., using poliv) or not.

RESPONSE: To address this comment, we now examined the response of stem cell and niche specific genes to SNP treatment in wt and the *nrpd1* mutants. Among the 12 stem cell- and niche-specific NO responsive genes in the wild type, none of them lost their response to SNP treatment (Figure. 1 for reviewer), indicating the NO mediated stem cell regulation via *AGO4-WUS* pathway does not rely on Pol V activity.

Figure. 1 for reviewer. NRPD1B does not mediate NO signaling in the shoot meristem.

a, b, RT-qPCR detected gene expression level of stem cell- and niche-specific genes in the wild type (a) and *nrpd1b-11* (b) mutants with and without SNP treatment. The data are shown as mean \pm s.d.; n=3 biological replicates, two-tailed Student's t-tests, **p<0.01, ***p<0.001.

To show that AGO4 is recruited to the same sites as WUS (e.g., to promoters of *NIA1* or *ARR7*) by chip-pcr would be affirmative.

RESPONSE: As suggested, we performed ChIP-PCR on AGO4 and found that AGO4 directly binds to the promoter of *ARR7*, in one of the promoter regions also bound by WUS. These results are now shown in the new (Supplementary Fig. 20).

Also, the promoters of potential target genes should be analyzed if they carry transposon fragments as target sites for RdDM.

They should also investigate the existence of small RNAs at the promoter regions of common

AGO4/WUS targets.

RESPONSE: We have carefully analyzed our new WUS ChIP-Seq data for the overlap between WUS chromatin binding and transposons and found that a substantial fraction of WUS binding events are located in these elements (Supplementary Fig. 14). This observation suggests that WUS may be involved in protecting stem cells from the activity of TEs. Conversely, we did not find evidence of TE sequences enriched among in the overlap of WUS and AGO4 target genes, suggesting that the chromatin binding likely does not involve small RNAs.

Scanning for siRNAs matches in the promoter regions of AGO4 and WUS targets, we identified about 51,256 and 20,595 siRNAs , respectively. However, only 354 (0.6%) of them were represented in both list (Figure. 2 for reviewer).

Figure. 2 for reviewer. Distribution pattern of siRNAs in the promoter region of AGO4/WUS common targets. Venn diagram showing the overlap of siRNAs in the promoter region of AGO4 and WUS targets.

Less importantly, it would also be interesting to investigate a potential connection to 22nt small RNAs, as there is a strong association between the silencing of NIA1 and NIA2 and the activity of small RNAs (Wu et al., Nature 581, 2020).

RESPONSE: As suggested, we performed small RNA sequencing on *ap1/cal* shoot apices with and without SNP treatment to test the small RNA behavior in response to NO and found that the majority of small RNAs were not significantly affected by increasing endogenous NO level (Figure. 3 for reviewer), indicating that NO does not have effects on small RNAs activity in the shoot meristem.

Figure. 3 for reviewer. Profiling of small RNAs in *ap1/cal* with and without SNP treatment. Analysis of small RNAs in *ap1/cal* shoot apices with and without SNP treatment using small RNA sequencing. Up indicates up-regulated upon SNP treatment, Down indicates down-regulated after SNP treatment, Normal indicates no significant changes.

Another concern is the analysis and interpretation of the DNA methylation data. First, AGO4 is almost entirely responsible for RdDM-mediated CHH methylation (Stroud et al., 2013). Still, many effects the authors claim to be connected to RdDM are visible only in *ago4/ago6/ago9*. Also, the choice of *drm1/drm2/met1* is unfortunate, as *met1* has global DNA methylation defects, not limited to RdDM. A better option would have been strict RdDM mutants e.g., *drm1/drm2*, *rdr2*, *rdm1*, or *nrpe1*. Could it be that the observed transcriptional changes are not related to RdDM but to additional functions of AGO6 and AGO9 or MET1? The authors should at least use one more RdDM mutant to reduce this concern.

It would also be excellent to see potential changes in promoter CHH methylation of NIA1 and NIA2 plus SNP.

RESPONSE: We thank the reviewer for the very constructive suggestion. We examined the effects by using additional RdDM mutants, such as *drm1/drm2* and *rdm1* and observed similar results found in the *ago4/ago6/ago9* and *drm1/drm2/met1* triple mutants. We have added this new data in a number of new supplementary figures. As suggested, we also analyzed the CHH methylation of NIA1 and NIA2 with SNP treatment, but did not observe any changes (Supplementary Table. 5).

Minor Comments:

Paragraph "Nitric oxide promotes peripheral zone fate via repression of WUS"

It is hard to see in Supplementary Fig. 1a the claim that NO accumulated to higher levels in the PZ compared to the CZ. Please indicate fluorescence intensities across the picture and show more examples of meristems.

RESPONSE: We have now examined more samples and carefully quantified the DAF-2DA fluorescent intensity (Supplementary Fig. 1).

Supplementary Fig. 2: the Signal for NIA2 is challenging to see. Given the difficulties of producing clear *in situ*, please provide at least antisense and/or knock-out mutant controls for all three genes.

RESPONSE: We thank the reviewer's suggestion. We agree with the reviewer's comment that the *in situ* picture of *NIA2* was not clear enough. We replaced the old image and included sense probes as control for all three NO genes in the revised manuscript.

To evaluate the functionality of all three genes: Did the authors observe phenotypic differences to wt in any of the double mutant combinations? For example, do *nia1/nia2/noa1* plants grow slower, or are they also producing fewer organs (do they flower with the same or a different number of leaves)? Are they growing normally on soil, or need additional nitrogen sources?

RESPONSE: Thank you for bringing up these important points. We have now carefully performed phenotypic analysis of the mutants and found that the emergence of the first true leaves is severely compromised in the mutants compared with wild type plants, indicating that the meristem activity is reduced in the mutants. We have added this new data in the revised manuscript.

It looks like the experiment in Supplementary Figure 1a-d has been done only with one replica. Either clarify or provide more biological replicas. It would be beneficial to show this experiment performed with 35S::ALCR;ALA::GUS to exclude the influence of ethanol on the staining.

RESPONSE: Thank you for the suggestion, we have examined more samples as well as including a control by using 35S::AlcR; AlcA::GUS control plants. We included this new data in the new Supplementary Fig 1.

Fig1 m,n): On which data is the quantification based? Is it linear measurements of *in-situ* hybridizations? Please clarify.

RESPONSE: We appreciate the reviewer's comment and apologized for the missing information. We have now added this information in the revised manuscript.

Supplementary Fig.5 comes right after supplementary Fig.2. Please make the order chronological to make reading easier.

RESPONSE: We thank the reviewer for the suggestion, and now it is fixed.

More importantly, Supplementary Fig.5 is an important control experiment to exclude a non-specific effect of NO on WUS expression. But it is not possible to judge from the images the validity. Fluorescence intensities from several meristems need to be quantified.

RESPONSE: We thank the reviewer for the suggestion. We have added the quantification data in the revised manuscript.

The line "We inspected our resources on direct WUS target genes". Which resources?

RESPONSE: : Thank you for pointing this out, we cited the reference in the revised manuscript.

RNA-seq data: please describe briefly in the text how DEGs are defined (which statistical test was used, what is the TPM/padj/foldchange cutoff?)

RESPONSE: We have added this information in the Methods and Supplementary Tables.

Fig.3: a: it is confusing to have both SNP/Mock down and SNP/Mock up in the same Venn diagram. Either show only the down or split them, or use upset plots.

RESPONSE: As suggested, we now show only the down list in the revised figure.

AGO4 is one of 10 AGO proteins most highly expressed in the SAM. Therefore it would be essential to show the specificity of their probe by using knock-out *ago4* plants.

RESPONSE: Following this reviewer's advice, we examined the *AGO4* expression pattern in *ago4* mutants by in situ hybridization, and found the signal was reduced to barely above background levels in the *ago4* mutant compared to the wild type, indicating the observed signal indeed is specific to AGO4 mRNA. In addition, we also included a negative control using the AGO4sense probe. This information is included in the new Supplementary Fig. 10.

Figure 3o: please indicate the SNP concentration.

RESPONSE: Thank you very much for pointing this out. We have added this information to Fig. 3o.

The authors report that AGO4 accumulates in the cytoplasm upon SNP treatment. Are the S-nitrosylation sites on a potential NLS or NES?

RESPONSE: It would indeed be interesting to investigate whether the S-nitrosylation sites of AGO4 occur on potential NLS or NES, however, to address this would require a thorough structure function analysis of AGO4, which is beyond the scope of this work.

Chapter "AGO4 protein connects WUS function with the RdDM pathway"

Sentence: "together with its orthologues AGO6 and AGO9. The functional relevance of AGO9 for RdDM has never been demonstrated.

As mentioned already above, the usage of *drm1/drm2/met1* was not a good choice. MET1 is a CG maintenance methyltransferase, and loss of MET1 has all kinds of secondary effects (e.g., late flowering). It would have been better to use other RdDM mutants like *drm1/drm1*, *rdr2*, or *poliv*.

RESPONSE: As pointed out above, we observed the similar effects by using additional RdDM mutants, we added the new data in the revised manuscript. AGO6 and AGO9 have been shown to be by and large biochemically equivalent of AGO4 (Havercker et al. Plant Cell 2010) and thus the redundant activity does not appear too surprising. This does not imply that AGO6 or AGO9 execute these functions in the wild type setting.

Figure 5a: What is the p-Value of the overlap (e.g., use a hypergeometric test). The reference for Zheng et al. should be in the text and the reference list.

RESPONSE: We have added the p-Value of the overlap in Fig. 5a, and cited the Zheng et al. .

Figure 5c: Please indicate the antibody used for IP in the Figure.

The IP should be repeated using *Arabidopsis* stably transformed *Arabidopsis* plants.

RESPONSE: As suggested, we have now carried out the IP in stable transgenic lines of *Arabidopsis*, demonstrating that endogenous AGO4 bind to a tagged form of WUS.

Which *ago4* allele was used? Supplementary Fig. 7: please write which mutant is in which genetic background. It would also be important to show the SAM size of Ler and Col wt plants. I assume *ago4wus7* plants are from an F2 population from the *ago4* x *wus7* cross. Are the LerXCcol plants also F2? LerXCcol f2 plants are problematic as they are very heterogeneous. The better experiment would have been to generate a CRIPR *ago4* knock-out in the *wus7* background.

RESPONSE: We thank the reviewer for bringing up this important point. We have added the missing information in the revised manuscript. In addition, we also examined the SAM sizes of Ler and Col wild type and added this new data in the revised manuscript.

Line " this phenotype was largely suppressed in *drm1/drm2/met1* triple mutant": please refer to

picture. Is it Supplementary Fig. 8 c/d?

RESPONSE: Thank you for bringing up this important point. We have added this information in the revised text.

Unfortunately, Supplementary Fig. 8 is also difficult to interpret as Met1 is not involved in RdDM but DRM1/DRM2, meaning the conclusion that "the RdDM pathway is required for WUS activity and proper SAM function" is not supported by the data. They could repeat this experiment with other RdDM mutants (as suggested above). It is also unclear to me how the authors interpret Supplementary Fig.8: Does that mean DNA methylation is required to activate CLV3? That would be in contrast to the fact that most DNA methylation mutants develop similarly to wt.

RESPONSE: As suggested, we have now investigated *drm1/drm2* and *rdm1* mutants, and observed fewer flower buds and reduced *WUS* expression (Supplementary Fig. 12), suggesting RdDM pathway is required for proper meristem activity.

Chapter: "Nitric oxide controls expression of key meristem regulators via DNA methylation"

The DNA methylation data need to be analyzed more carefully. Which treatment length of SNP was used for Bisulfite sequencing? It is critical to distinguish between CG, CHG, and CHH methylation, as only CHH methylation is affected by the loss of AGO4. What kind of DNA methylation levels do the authors describe? Weighted DNA methylation? It is unclear to me how Fig. 6 b shows a trend towards "reduced DNA methylation"?

RESPONSE: We thank the reviewer for bringing up these important points. We have now carefully analyzed the EM-seq by separating CG, CHG and CHH methylations, and added new information in the revised manuscript.

Supplementary Fig.9: I need clarification on why the authors show total DNA methylation levels, as this is irrelevant to AGO4 function. They should distinguish CG, CHG, CHH methylation, and promoter and gene body methylation. It is also unclear to me how "total 5-mC" was calculated, as it is unclear from the description in M&M. Please use methylation levels as defined in Schultz et al., Trends Genet 2012., or describe better how the calculation was done.

RESPONSE: Following the reviewer's advice, we have now included information on CG, CHG, and CHH methylation in the revised manuscript also added the information for how we calculate the total 5-mC in the M&M part.

To ease the reproducibility of the study, please include the atg numbers plus Stock-identifiers for all genes and mutants used in the study.

RESPONSE: Following the reviewer's suggestion, we have provided this information in the revised manuscript.

Discussion:

I do not know about RdDM activity in animals; the papers they refer to do not show this.

RESPONSE: We appreciate the reviewer's comments. We have added new references to the revised manuscript that show the comparison of RdDM activity both in animals and plants.

Methods: Please provide the C to T conversion rates (using the chloroplast genome) and genome coverage for the EM-seq data.

RESPONSE: Thank you for the suggestion. We have added this information in the revised manuscript.

--Reviewer #3

The manuscript by Zang et al. entitled "Nitric Oxide controls shoot meristem activity via regulation of DNA methylation" reports the roles of NO and RdRM in the SAM. AGO4 is inhibited by NO at the transcriptional level and also post-transcriptional level, which is S-nitrosylation dependent. AGO4 interacts with WUS at the protein level, and this interaction is disrupted by NO treatment. The roles of RdRM in the SAM have been reported. For example, rice RdRM mutant plants have prominent SAM defects. This study links NO regulation to the roles of RdRM and WUS. Together, this study provides many interesting clues but there seems to be insufficient support for the models proposed by the authors. In the model, many links are not well established either. Below are my specific comments.

RESPONSE: We thank the reviewer for pointing out the interest in our study and helping us to address the limitations.

1. How SAM size, in particular PZ size was determined? UFO has been used as a PZ marker (Lee et al., 1997 Curr Biol doi: 10.1016/s0960-9822(06)00053-4). A recent work shows AS2 expression in the PZ as well (Burian et al., 2022 Nat Plants doi: 10.1038/s41477-022-01111-3). Or KAN1 highlighting the outer boundary of the SAM.

RESPONSE: SAM size was quantified by measuring the area below the L1 cell layer and the max distance between primordia. Following the reviewer's suggestion, we have now examined the expression patterns of *UFO* as a marker for PZ and *CUC2* as a marker for boundaries in NO biosynthesis and DNA methylation mutants. The results fully support our conclusions and have are shown in the new Supplementary Fig 4.

2. Fig. 2a-c, 3 h treatment w/ SNP or cPTIO causes WUS signal changes. Was the pWUS::2xVenusNLS reporter used? If so, the reduction after SNP treatment seems suggest a fast degradation of 2xVenusNLS. Why Venus is so unstable? Note that the authors have shown that GFP fluorescence is not responsive to NO levels (Suppl Fig 5a-c). Venus is a fast maturing protein, but 3 h is still short for a substantial portion of Venus to mature (Balleza et al, 2018 Nat Methods doi: 10.1038/nmeth.4509).

RESPONSE: Indeed, the reduction of the WUSp::2xVenusNLS reported matched the dramatic reduction in *WUS* transcripts we observed after 3h SNP treatments both in RT-qPCR and *in situ* experiments (Fig. 2 g, h and n).

3. The protein interaction between WUS and AGO4 is interesting. However, the biological relevance is less clear. If the interaction contributes to WUS activity as a transcription factor, differential expression of WUS-target genes should be identified in *ago4* mutant plants. Alternatively, if WUS participates in the RdRM activity of AGO4, AGO4 targets should be different in *wus* mutants or WUS-OX lines.

RESPONSE: We agree that the interaction of WUS and AGO4 is interesting and that the underlying mechanisms may be complex. As described in the response to the other reviewers, we have now analyzed the potential connection of WUS to the RdDM pathway output, but did neither find evidence for transposon fragment in WUS response genes, nor overlap in the small RNA signature in chromatin binding regions of WUS and AGO4. In contrast, we did find that WUS bind to many transposable elements, suggesting that it may contribute to stem cell protection against mobile genetic elements.

4. The DAF-2DA staining results is not very clear. Is the signal enriched in incipient primordia or the boundaries between primordia?

RESPONSE: After carefully checking more samples for DAF-2DA staining, we found an DAF-2DA fluorescence gradient from boundaries to the center, with the signal is tending to be enriched in the boundaries. We have now included careful quantification in the new Supplementary Fig. 1.

5. The expression of both WUS and CLV3 are enhanced in the *nia* triple mutant. Why?

RESPONSE: Currently, we cannot provide a mechanism for the repression of WUS by NO.

6. In Fig. 2j-l, CLV3 signal seems to be enhanced after SNP and cPTIO treatment.

RESPONSE: The CLV3 expression with and SNP and cPTIO treatments lies within the levels of experimental variation for all assays tested, be it fluorescent reporter, *in situ* hybridization, or RT-

qPCR.

7. To really test whether S-nitrosylation is responsible for AGO4-WUS interaction, the authors need to disrupt all possible S-nitrosylation sites, which can be identified by mass spectrum analysis.

RESPONSE: We thank the reviewer for the suggestion. We fully agree with the reviewer that an identification of all S-nitrosylation sites of AGO4 would advance our insights into AGO4 function including its interaction with WUS. However, this ambitious project appears beyond the scope of our study.

REVIEWERS' COMMENTS

Reviewer #1 (Remarks to the Author):

This reviewer has no more questions about the revised manuscript.

Reviewer #2 (Remarks to the Author):

The authors addressed all my concerns. Some questions remain open about the exact mechanisms. However, the authors showcase an intriguing connection of the RdDM pathway with meristem functioning, and between AGO4, WUS, and NO signaling, justifying publishing the work in a general journal like Nature Communications.

Reviewer #3 (Remarks to the Author):

This is a revision of a previously reviewed paper, Zeng et al., 'Nitric Oxide controls shoot meristem activity via regulation of DNA methylation'. Whereas most of my concerns have been addressed, One important one remains.

(Previous) 2. Fig. 2a-c, 3 h treatment w/ SNP or cPTIO causes WUS signal changes. Was the pWUS::2xVenusNLS reporter used? If so, the reduction after SNP treatment seems suggest a fast degradation of 2xVenusNLS. Why Venus is so unstable? Note that the authors have shown that GFP fluorescence is not responsive to NO levels (Suppl Fig 5a-c). Venus is a fast maturing protein, but 3 h is still short for a substantial portion of Venus to mature (Balleza et al, 2018 Nat Methods doi: 10.1038/nmeth.4509).

RESPONSE: Indeed, the reduction of the WUSp::2xVenusNLS reported matched the dramatic reduction in WUS transcripts we observed after 3h SNP treatments both in RT-qPCR and in situ experiments (Fig. 2 g, h and n).

I do not understand why GFP and Venus signals were reduced in 3 h after SNP or cPTIO treatment. Note that GFP per se is not sensitive to NO levels as shown by the authors. Is the degradation of GFP/Venus extremely fast in stem cells? The half life of GFP/Venus in mammalian cells is ~26 h (Corish and Tyler Smith, 1999 PNAS doi: 10.1093/protein/12.12.1035)

In addition, the DAF-2DA stain result is key to the paper, but remains of poor quality. Better annotation is needed to orient readers through Suppl Fig. 1. It is unclear where are PZ and CZ in Suppl. Fig. 1a-f.

Point-by-point response to the reviewer's comments

We appreciate all the comments raised by editor and all reviewers. We have revised the text and supplement accordingly and specific responses to the reviewer's comments can be found below:

--Reviewer #1 (Remarks to the Author):

This reviewer has no more questions about the revised manuscript.

RESPONSE: We express our gratitude to the reviewer once again for providing thorough feedback and valuable input that have significantly enhanced the quality of our manuscript.

Reviewer #2 (Remarks to the Author):

The authors addressed all my concerns. Some questions remain open about the exact mechanisms. However, the authors showcase an intriguing connection of the RdDM pathway with meristem functioning, and between AGO4, WUS, and NO signaling, justifying publishing the work in a general journal like Nature Communications.

RESPONSE: We thank the reviewer for recognizing the significance of our work and the intriguing connections we've explored. We greatly appreciate your positive feedback and support for publishing in Nature Communications.

Reviewer #3 (Remarks to the Author):

This is a revision of a previously reviewed paper, Zeng et al., 'Nitric Oxide controls shoot meristem activity via regulation of DNA methylation'. Whereas most of my concerns have been addressed, One important one remains.

(Previous) 2. Fig. 2a-c, 3 h treatment w/ SNP or cPTIO causes WUS signal changes. Was the pWUS::2xVenusNLS reporter used? If so, the reduction after SNP treatment seems suggest a fast degradation of 2xVenusNLS. Why Venus is so unstable? Note that the authors have shown that GFP fluorescence is not responsive to NO levels (Suppl Fig 5a-c). Venus is a fast maturing protein, but 3 h is still short for a substantial portion of Venus to mature (Balleza et al, 2018 Nat Methods doi: 10.1038/nmeth.4509).

RESPONSE: Indeed, the reduction of the WUSp::2xVenusNLS reported matched the dramatic reduction in WUS transcripts we observed after 3h SNP treatments both in RT-qPCR and in situ experiments (Fig. 2 g, h and n).

I do not understand why GFP and Venus signals were reduced in 3 h after SNP or cPTIO

treatment. Note that GFP per se is not sensitive to NO levels as shown by the authors. Is the degradation of GFP/Venus extremely fast in stem cells? The half life of GFP/Venus in mammalian cells is ~26 h (Corish and Tyler Smith, 1999 PEDS doi: 10.1093/protein/12.12.1035)

RESPONSE: We appreciate the reviewer's additional comments. Despite Corish et. al, 1999 report indicating a 26 h half-life of GFP/Venus in mammalian cells, it has been observed multiple times independently that GFP/Venus fluorescence can be reduced within as little as 1 to 24 hours under different conditions or treatments in plant cells (Müller et al 2008, Hazak et al 2014, Xuan et al 2016, Wang et al 2016, Hong et al 2017, Rast-Somssich et al 2017 and Bennett et al 2014), suggesting a potential difference in GFP/Venus half-life between mammalian and plant cells. Supported by parallel evidences from our RT-qPCR and in situ experiments, we are confident in the robustness of our findings regarding the reduction of GFP/Venus signals in WUSp::2xVenusNLS plants under SNP treatments.

In addition, the DAF-2DA stain result is key to the paper, but remains of poor quality. Better annotation is needed to orient readers through Suppl Fig. 1. It is unclear where are PZ and CZ in Suppl. Fig. 1a-f.

RESPONSE: We appreciate the reviewer comments. As suggested, we have marked the CZ and PZ region accordingly, to make them more clear.